# Predictive learning rules generate a cortical-like replay of probabilistic sensory experiences

**Toshitake Asabuki[1,2,3]\*, Tomoki Fukai[1]\***

[1]Okinawa Institute of Science and Technology Graduate University, Okinawa, Japan; [2]RIKEN Center for Brain Science, RIKEN ECL Research Unit, Wako, Japan; [3]RIKEN Pioneering Research Institute (PRI), Wako, Japan

## eLife Assessment

This **valuable** study investigates how biologically plausible learning mechanisms can support assembly formation that encodes statistics of the environment, by enabling neural sampling that is based on within-assembly connectivity strength. It **convincingly** shows that assembly formation can emerge from predictive plasticity in excitatory synapses, while two types of plasticity in inhibitory synapses are required: inhibitory homeostatic (predictive) plasticity and inhibitory competitive (anti-predictive) plasticity.

**\*For correspondence:**
toshitake.asabuki@gmail.com (TA);
tomoki.fukai@oist.jp (TF)

**Competing interest:** The authors declare that no competing interests exist.

**Abstract** The brain is thought to construct an optimal internal model representing the probabilistic structure of the environment accurately. Evidence suggests that spontaneous brain activity gives such a model by cycling through activity patterns evoked by previous sensory experiences with the experienced probabilities. The brain's spontaneous activity emerges from internally driven neural population dynamics. However, how cortical neural networks encode internal models into spontaneous activity is poorly understood. Recent computational and experimental studies suggest that a cortical neuron can implement complex computations, including predictive responses, through soma–dendrite interactions. Here, we show that a recurrent network of spiking neurons subject to the same predictive learning principle provides a novel mechanism to learn the spontaneous replay of probabilistic sensory experiences. In this network, the learning rules minimize probability mismatches between stimulus-evoked and internally driven activities in all excitatory and inhibitory neurons. This learning paradigm generates stimulus-specific cell assemblies that internally remember their activation probabilities using within-assembly recurrent connections. Our model contrasts previous models that encode the statistical structure of sensory experiences into Markovian transition patterns among cell assemblies. We demonstrate that the spontaneous activity of our model well replicates the behavioral biases of monkeys performing perceptual decision making. Our results suggest that interactions between intracellular processes and recurrent network dynamics are more crucial for learning cognitive behaviors than previously thought.

## Introduction

The brain is believed to construct an internal statistical model of an uncertain environment from sensory information streams for predicting the external events that are likely to occur. Evidence suggests that spontaneous brain activity learns the representation of such a model through repeated experiences of sensory events. In the cat visual cortex, spontaneously emerging activity patterns cycle through cortical states that include neural response patterns to oriented bars (*Kenet et al., 2003*). In

the ferret visual cortex, spontaneous activity gradually resembles a superposition of activity patterns evoked by natural scenes, eventually giving an optimal model of the visual experience (*Berkes et al., 2011*). As replay activities can provide prior information for hierarchical Bayesian computation by the brain (*Ernst and Banks, 2002*; *Körding and Wolpert, 2004*; *Friston, 2010*; *Fiser et al., 2010*; *Bastos et al., 2012*; *Orbán et al., 2016*; *Legaspi and Toyoizumi, 2019*), clarifying how the brain learns the spontaneous replay of optimal internal models is crucial for understanding whole-brain computing. However, the neural mechanisms underlying this modeling process are only poorly understood.

Several mechanisms of the brain's probabilistic computation have been explored (*Jimenez Rezende and Gerstner, 2014*; *Li et al., 2022*). Models with reverberating activity are particularly interesting owing to their potential ability to generate spontaneous activity. For instance, spiking neuron networks with symmetric recurrent connections were proposed for Markov Chain Monte Carlo sampling of stochastic events (*Buesing et al., 2011*; *Bill et al., 2015*). Spike-timing-dependent plasticity (STDP) was used to organize spontaneous sequential activity patterns, providing a predictive model of sequence input (*Hartmann et al., 2015*). However, previous models did not clarify how recurrent neural networks learn the spontaneous replay of the probabilistic structure of sensory experiences, for which these networks should learn the accurate probabilities of sensory stimuli and an appropriate excitation–inhibition balance simultaneously. Moreover, previous models assumed that each statistically salient stimulus in temporal input is already segregated and is delivered to a pre-assigned assembly of coding neurons, implying that the recurrent network, at least partly, knows the stochastic events to be modeled before learning. How the brain extracts salient events for statistical modeling has not been addressed.

Here, we present a learning principle to encode the experiences' probability structure into spontaneous network activity. To this end, we extensively use the synaptic plasticity rule proposed previously based on the hypothesis that the dendrites of a cortical neuron learn to predict its somatic responses (*Urbanczik and Senn, 2014*; *Asabuki and Fukai, 2020*). We generalize the hypothetical predictive learning to a learning principle at the entire network level. Namely, in a recurrent network driven by external input, we ask all synapses on the dendrites of each excitatory or inhibitory neuron to learn to predict its somatic responses (although the dendrites will not be explicitly modeled). This enables the network model to simultaneously learn the events' probabilistic structure and the excitation–inhibition balance required to replay this structure. Further, our network model requires no pre-assigned cell assemblies since the model neuron can automatically segment statistically salient events in temporal input (*Asabuki and Fukai, 2020*) – a cognitive process known as 'chunking' (*Fujii and Graybiel, 2003*; *Jin and Costa, 2010*; *Jin et al., 2014*; *Schapiro et al., 2013*; *Zacks et al., 2001*). Intriguingly, the cell assemblies generated by our model store their replay probabilities primarily in the within-assembly network structure, and intrinsic dynamical properties of membership neurons also contribute to this coding. This is in striking contrast to other network models that encode probabilities into the Markovian transition dynamics among cell assemblies (*Buesing et al., 2011*; *Hartmann et al., 2015*; *Asabuki and Clopath, 2024*).

Our model trained on a perceptual decision-making task can replicate both unbiased and biased decision behaviors of monkeys without fine-tuning of parameters (*Hanks et al., 2011*). In addition, in a network model consisting of distinct excitatory and inhibitory neural populations, our learning rule predicts the emergence of two types of inhibitory connections with different computational roles. We show that the emergence of the two inhibitory connection types is crucial for robust learning of an optimal internal model.

## Results
### Replay of probabilistic sensory experiences – a toy example
We first explain the task our model solves with a toy example. Consider a task in which the animal should decide whether a given stimulus coincides with or resembles any of two previously learned stimuli. Whether the animal learned these stimuli with a 50–50 chance or a 30–70 chance should affect the animal's anticipation of their occurrence and hence affect its decision.

It has been suggested that spontaneous activity expresses an optimal internal model of the sensory environment (*Berkes et al., 2011*). In our toy example, the evoked activity patterns of the two stimuli

should be spontaneously replayed with the same probabilities as these stimuli were experienced during learning:

$$\left\langle P\left(\text{features}|\text{input, model}\right)\right\rangle_{P(\text{input})} = P\left(\text{features}|\text{model}\right),$$

where features = {stimulus 1, stimulus 2} and the right-hand side expresses the probabilities of replayed activities. The angular brackets indicate averaging over the stimuli. According to Hebb's hypothesis, two cell assemblies should be formed to memorize the two stimuli in the toy example. Moreover, the spontaneous replay of these cell assemblies should represent the probabilities given in the right-hand side of the above equation. Below, we propose a mathematical principle of learning to achieve these requirements.

## Prediction-driven synaptic plasticity for encoding an internal model

We previously proposed a learning rule for a single two-compartment neuron (*Asabuki and Fukai, 2020*). Briefly, our previous model learns statistically salient features repeated in input sequences by

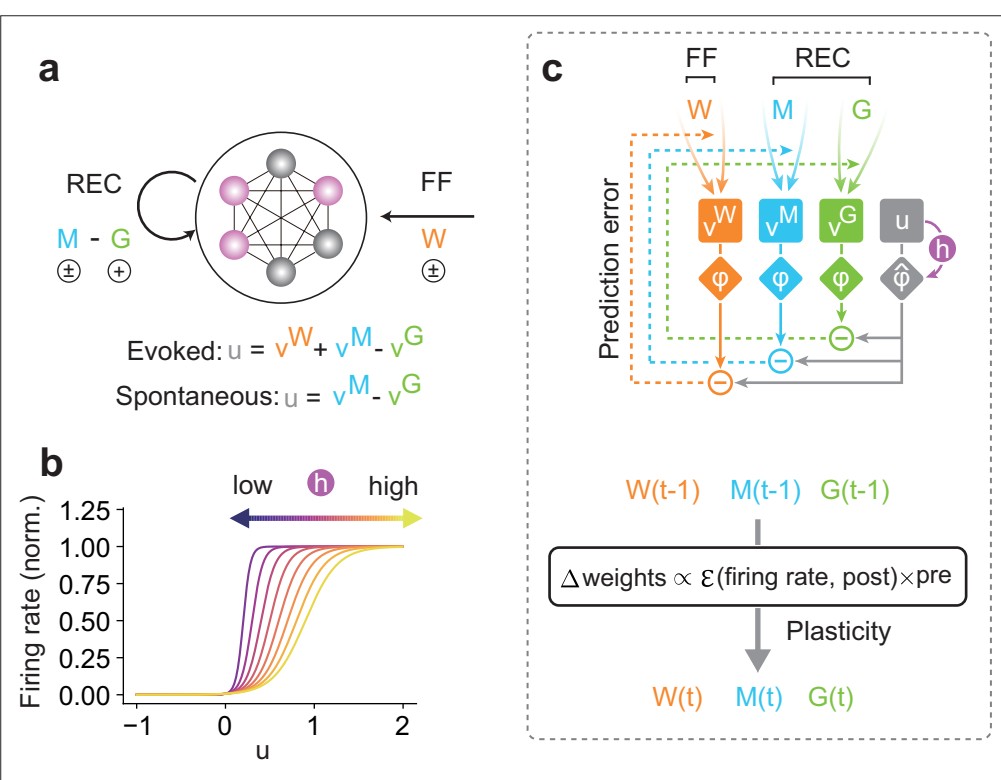

**Figure 1.** Unsupervised prior learning in a recurrent neural network. (**a**) A schematic of a network model is shown. The interconnected circles denote the model neurons, of which the activities are controlled by two types of inputs: feedforward (FF) and recurrent (REC) inputs. Colored circles indicate active neurons. Here, $v^W$ denote FF, and $v^M$ denote REC connections. We considered two modes of activity (i.e., evoked and spontaneous activity). In the evoked mode, the membrane potential u of a network neuron was calculated as a linear combination of inputs across all different connections ($v^W$, $v^M$, and $v^G$). This evoked mode is considered during the learning phase, when all synapses attempt to predict the network activity, as we will explain in the main text. Once all synapses are sufficiently learned, all FF inputs are removed, and the network is driven spontaneously (spontaneous mode). Our interest lies in the statistical similarity of the network activity in these two modes. (**b**) The gain and threshold of output response function was controlled by a dynamic variable, h, which tracks the history of the membrane potential. (**c**) A schematic of the learning rule for a network neuron is shown (top). During learning, for each type of connection on a postsynaptic neuron, synaptic plasticity minimizes the error between output (gray diamond) and synaptic prediction (colored diamonds). Note that all types of synapses share the common plasticity rule, where weight updates are calculated as the multiplication of the error term and the presynaptic activities (bottom). Our hypothesis is that such plasticity rule allows a recurrent neural network to spontaneously replay the learned stochastic activity patterns without external input.

minimizing the error between somatic and dendritic response probabilities without external supervision to identify the temporal locations of these features. In this study, we extend this plasticity rule to recurrent networks by asking all neurons in a network to minimize the error in response probabilities between the internally generated and stimulus-evoked activities (*Figure 1*). Our central interest is whether this learning principle generates spontaneous activity representing the statistical model of previous experiences.

We first introduce our learning principle using a recurrent network model (nDL model) that does not obey Dale's law for distinguishing between excitatory and inhibitory neurons (Materials and methods). A more realistic model with distinct excitatory and inhibitory neuron pools will be shown later. The nDL model consists of Poisson spiking neurons, each receiving Poisson spike trains from all input neurons via a modifiable all-to-all afferent feedforward connection matrix $W$ (*Figure 1a*). These input neurons may be grouped into multiple input neuron groups responding to different sensory features. Due to the all-to-all connectivity, the afferent input has no specific predefined structure. Two types of all-to-all modifiable recurrent connections, $M$ and $G$, exist among the neurons. Matrix $M$ is a mixture of excitatory and inhibitory connections, and matrix $G$ represents inhibitory-only connections. Due to a minus sign for $v^G$, all components of $G$ are positive. The firing rate of neurons is defined as a modifiable sigmoidal function of the membrane potential (*Figure 1b*), which we will explain later in detail. All types of connections, both afferent and recurrent ones, are modifiable by unsupervised learning rules derived from a common principle: on each neuron, all synapses learn to predict the neuron's response optimally (*Figure 1c*: see Materials and methods). In reality, all synaptic inputs may be terminated on the dendrites, although they are not modeled explicitly.

Without a teaching signal, predictive learning may suffer a trivial solution problem in which all synapses vanish, and hence all neurons become silent (*Asabuki and Fukai, 2020*). To avoid it, we homeostatically regulate the dynamic range of each neuron (i.e., the slope and threshold of the response function) according to the history $h$ of its subthreshold activity (see *Equations 6–8*). When the value of $h$ is increased, the neuron's excitability is lowered (*Figure 1b*). The input–output curves of neurons are known to undergo homeostatic regulations through various mechanisms (*Chance et al., 2002*; *Mitchell and Silver, 2003*; *Torres-Torrelo et al., 2014*). Though no direct experimental evidence is available for our homeostatic process via $h$, it mathematically avoids saturating neuronal activity.

Note that the present homeostatic regulation of intrinsic excitability differs from the homeostatic synaptic scaling mechanism. The role of homeostatic synaptic scaling in generating irregular cell-assembly activity patterns was previously studied computationally (*Hiratani and Fukai, 2014*; *Litwin-Kumar and Doiron, 2014*; *Zenke et al., 2015*). However, unlike the present model, the previous models did not address whether and how synaptic scaling contributes to statistical modeling by recurrent neural networks. Furthermore, unlike our model, in which neurons in the recurrent layer and input neurons are initially connected in an all-to-all manner, most previous models assumed preconfigured receptive fields for recurrent-layer neurons, implying that these models had predefined stimulus-specific cell assemblies.

## Cell-assembly formation for learning statistically salient stimuli

We first explain how our network segments salient stimuli and forms stimulus-specific cell assemblies via network-wide predictive learning rules. To this end, we tested a simple case in which two non-overlapping input groups are intermittently and repeatedly activated with equal probabilities. The two input patterns were separated by irregular, low-frequency, unrepeated spike trains of all input neurons (Materials and methods). We will consider input patterns with unequal occurrence probabilities later. After several presentations of individual input patterns, each network neuron responded selectively to one of the repeated patterns (*Figure 2a*). This result is consistent with our previous results (*Asabuki and Fukai, 2020*) that the plasticity of feedforward connections segments input patterns. Indeed, feedforward synapses W on each neuron were strengthened or weakened when they mediated its preferred or non-preferred stimulus, respectively (*Figure 2b*, left; *Figure 2c*). Inhibitory connections G grew between neurons within the same assembly but not between assemblies (*Figure 2b*, right; *Figure 2c*, bottom), enhancing the decorrelation of within-assembly neural activities (*Asabuki and Fukai, 2020*). Recurrent connections M were modified to form stimulus-specific cell assemblies, as evidenced by the self-organization of excitatory (*Figure 2c*, top) and inhibitory (*Figure 2c*, bottom)

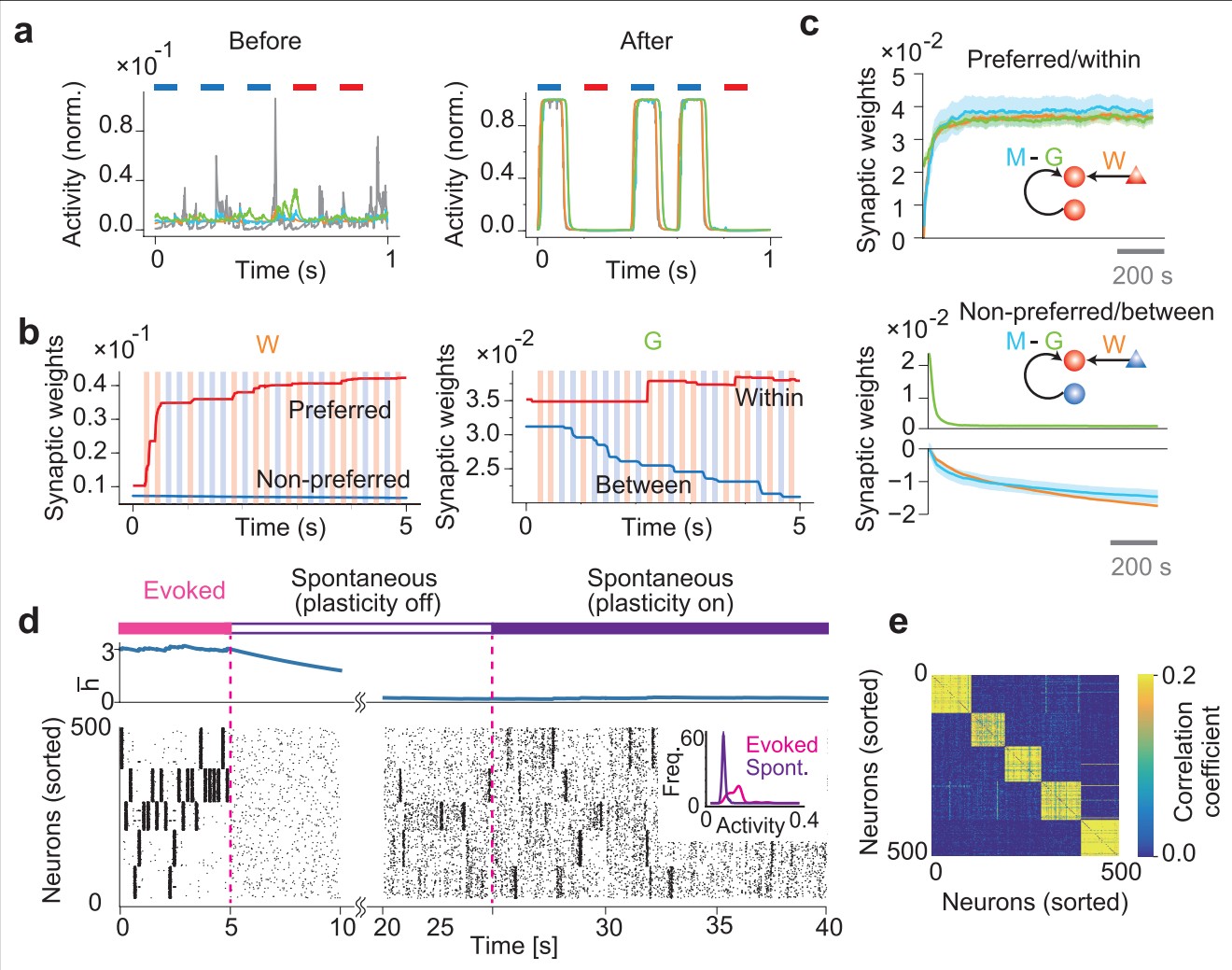

**Figure 2.** Formation of stimulus-selective assemblies in a recurrent network. (**a**) Example dynamics of neuronal output and synaptic predictions are shown before (left) and after (right) learning. Colored bars at the top of the figures represent periods of stimulus presentations. (**b**) Example dynamics of feedforward connection W and inhibitory connection G are shown. W-connections onto neurons organizing to encode the same or different input patterns are shown in red and blue, respectively. Similarly, the same colors are used to represent G connections within and between assemblies. (**c**) Dynamics of the mean connection strengths are shown on neuron in cell assembly 1. Shaded areas represent SDs over 10 samples. In the schematic, triangles indicate input neurons and circles indicate network neurons. The color of each neuron indicates the stimulus preference of each neuron. (**d**) Example dynamics of the averaged dynamical variable $\bar{h}$ (top) and the learned network activity (bottom) are shown. The dynamical variables are averaged over the entire network. Neurons are sorted according to their preferred stimuli. During the spontaneous activity, afferent inputs to the network were removed. The inset shows the firing rate distribution of the evoked and the spontaneous activity. (**e**) Correlation coefficients of spontaneous activities of every pair of neurons are shown.

recurrent connections within and between cell assemblies, respectively. The inhibitory components are necessary for suppressing the simultaneous replay of different cell assemblies, as shown later.

We then investigated whether and how spontaneous activity preserves and replays these cell assemblies in the absence of afferent input. To demonstrate this in a more complex task, we trained the network with afferent input involving five repeated patterns and then removed the input and observed post-training spontaneous network activity (*Figure 2d*). The termination of afferent input initially lowered the activities of neurons, but their dynamic ranges gradually recovered with the excitability of the neural population (indicated by the population-averaged $h$ value), and the network eventually started spontaneously replaying the learned cell assemblies. All plasticity rules were turned off during the recovery period (about 20 s from the input termination), after which the network settled in a stable spontaneous firing state (plasticity off), with firing rates lower than those of the

evoked activity (inset). Then, the plasticity rules could be turned on (plasticity on) without drastically destroying the structure of spontaneous replay. Intriguingly, spontaneous neuronal activities were highly correlated within each cell assembly but were uncorrelated between different cell assemblies (*Figure 2e*). This was because self-organized recurrent connections $M$ were excitatory within each cell assembly, whereas the between-assembly recurrent connections were inhibitory, as in *Figure 2c*.

Thus, the network model successfully segregates, remembers, and replays stimulus-evoked activity patterns in temporal input. The loss of between-assembly excitatory connections is interesting as it indicates that the present spontaneous reactivation is not due to the sequential activation of cell assemblies. This can also be seen from the relatively long intervals between consecutive cell-assembly activations: spontaneous neural activity does not propagate directly from one cell assembly to another (*Figure 2d*). Indeed, within-assembly excitation is the major cause of spontaneous replay in this model, which we will study later in detail.

In summary, we have proposed the predictive learning rules as a novel plasticity mechanism for all types of synapses (i.e., feedforward and recurrent connections). We have shown that the plasticity rules in our model learn the segmentation of salient patterns in input sequences and form pattern-specific cell assemblies without preconfigured structures. We also showed that our model replays the learned assemblies even when external inputs were removed.

## Replays of cell assemblies reflect a learned statistical model

We now turn to the central question of this study. We asked whether internally generated network dynamics through recurrent synapses (i.e., spontaneous replay of cell assemblies) can represent an optimal model of previous sensory experiences. Specifically, we examined whether the network spontaneously reactivates learned cell assemblies with relative frequencies proportional to the probabilities with which external stimuli activated these cell assemblies during learning. We addressed these questions in slightly more complex cases with increased numbers of external stimuli.

We first examined a case with five stimuli in which stimulus 1 was presented twice as often as the other four stimuli (*Figure 3a*). Hereafter, the probability ratio refers to the relative number of times stimulus 1 is presented during learning. For instance, the case shown in *Figure 2d* represents the probability ratio one. As in *Figure 2d*, the network self-organized five cell assemblies to encode stimuli 1–5 and replayed all of them in subsequent spontaneous activity (*Figure 3b*). We found that output neurons were activated more frequently and strongly in cell assembly 1 than in other cell assemblies. Therefore, we assessed quantitative differences in neuronal activity between different cell assemblies by varying the probability ratio. The neuronal firing rate of cell assembly 1 relative to other cell assemblies increased approximately linearly with an increase in the probability ratio (*Figure 3c*). Similarly, the size of cell assembly 1 relative to other cell assemblies also increased with the probability ratio (*Figure 3d*). However, neither the relative firing rate nor the relative assembly size faithfully reflects changes in the probability ratio: scaling the probability ratio with a multiplicative factor does not scale these quantities with this factor. Therefore, we further investigated whether the assembly activity ratio, the ratio in the total firing rate of cell assembly 1 to other cell assemblies (Materials and methods), scales faithfully with the probability ratio of cell assembly 1. This was the case: the scaling was surprisingly accurate (*Figure 3e*).

To examine the ability of the nDL network further, we trained it with five stimuli occurring with various probabilities (*Figure 3f* and *Figure 3—figure supplement 1a*). After learning, the spontaneous activity of the model replayed the learned cell assemblies at the desired ratios of population firing rates (*Figure 3g* and *Figure 3—figure supplement 1b*).

We then asked whether our model would learn a prior distribution for more stimuli. To this end, we presented seven stimulus patterns to the same network with graded probabilities (*Figure 3—figure supplement 1c*). The self-organized spontaneous activity exhibited cell assemblies that well learned the graded probability distribution of these stimuli (*Figure 3—figure supplement 1d*). These results demonstrate that the trained network remembers the probabilities of repetitively experienced stimuli by the spontaneous firing rates of the encoding cell assemblies and that this dynamical coding scheme has a certain degree of scalability.

So far, we have represented external stimuli with non-overlapping subgroups of input neurons. However, in biologically realistic situations, input neuron groups may share part of their membership neurons. We tested whether the proposed model could learn the probability structure of overlapping

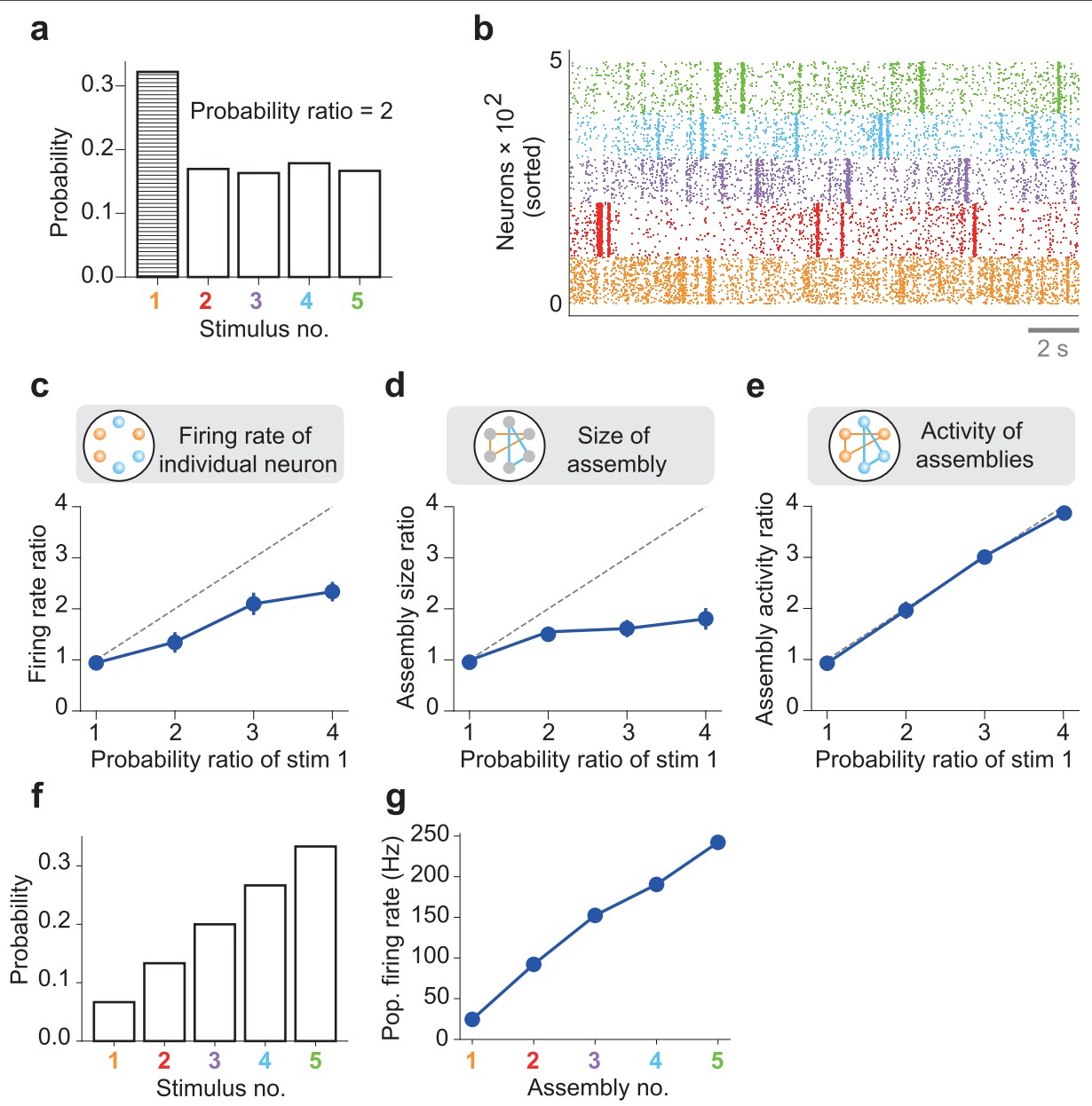

**Figure 3.** Priors coded in spontaneous activity. An nDL network was trained with five probabilistic inputs. (**a**) Stimulus 1 appeared twice as often as the other four stimuli during learning. The example empirical probabilities of the stimuli used for learning are shown. (**b**) The spontaneous activity of the trained network shows distinct assembly structures. (**c**) The mean ratio of the population-averaged firing rate of assembly 1 to those of the other assemblies is shown for different values of the occurrence probability of stimulus 1. Vertical bars show SDs over five trials. A diagonal dashed line is a ground truth. (**d**) Similarly, the mean ratios of the size of assembly 1 to those of the other assemblies are shown. (**e**) The mean ratios of the total activities of neurons in assembly 1 to those of the other assemblies are shown. (**f**) Five stimuli occurring with different probabilities were used for training the nDL model. (**g**) The population firing rates are shown for five self-organized cell assemblies encoding the stimulus probabilities shown in (**f**).

The online version of this article includes the following figure supplement(s) for figure 3:

**Figure supplement 1.** Prior encoding by the nDL model.

**Figure supplement 2.** Learning occurrence probabilities of overlapped input patterns.

input patterns in a case where two input neuron groups shared half of their members. The two patterns were presented with probabilities of 30% and 70%, respectively (**Figure 3—figure supplement 2a**). After sufficient learning, the network model generated two assemblies that encoded the two stimuli without sharing the coding neurons (**Figure 3—figure supplement 2b**) and replayed these

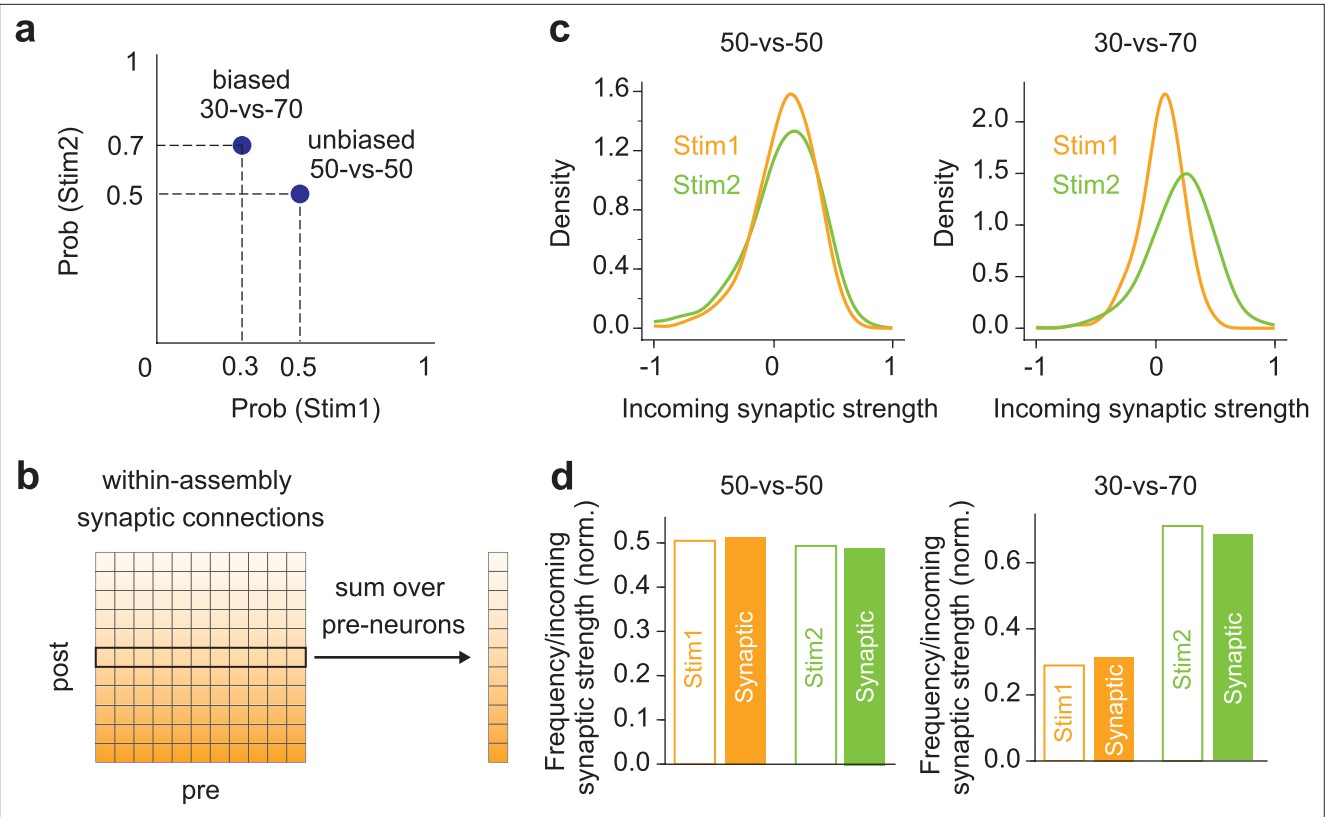

**Figure 4.** Probability encoding by learned within-assembly synapses. (**a**) Two input stimuli were presented in two protocols: uniform (50% vs. 50%) or biased (30% vs. 70%). (**b**) The total incoming synaptic strength on each neuron was calculated within each cell assembly. (**c**) *left*, The distributions of incoming synaptic strength are shown for the learned assemblies in the 50-vs-50 case. *right*, Same as in the left figure, but in the 30-vs-70 case. (**d**) *left*, The empirical probabilities of stimuli 1 and 2 and the normalized excitatory incoming weights within assemblies are compared in the 50-vs-50 case. *right*, Same as in the left figure, but in the 30-vs-70 case.

The online version of this article includes the following figure supplement(s) for figure 4:

**Figure supplement 1.** Within-assembly connections encode the probability structures.

**Figure supplement 2.** Inhibitory plasticity during learning is necessary to stabilize spontaneous activity.

**Figure supplement 3.** Crucial roles of inhibitory plasticity in prior learning.

**Figure supplement 4.** Distinct assembly replay after sequence.

**Figure supplement 5.** Role of dynamical variable $h$ in spontaneous replay of assemblies.

**Figure supplement 6.** Learning of multivariate priors with assemblies.

assemblies with frequencies proportional to the stimulus presentation probabilities (*Figure 3—figure supplement 2c*). The results look reasonable because each neuron in the network segments one of the stimulus patterns, and recurrent connections within each non-overlapping assembly can encode the probability of its replay.

Altogether, these results suggest that our model spontaneously replays learned cell assemblies with relative frequencies proportional to the probability that each cell assembly was activated during the learning phase. We have shown that the population activities of assemblies, rather than the firing rates of individual neurons, encode the occurrence probabilities of stimulus patterns.

## Within-assembly recurrent connections encode probabilistic sensory experiences

To understand the mechanism underlying the statistical similarity between the evoked patterns and spontaneous activity, we then investigated whether and how biases in probabilistic sensory experiences influence the strengths of recurrent connections. To this end, we compared two cases in which two input patterns (stim 1 and stim 2) occurred with equal (50% vs. 50%) and different (30% vs. 70%)

probabilities during learning (*Figure 4a*). From the results shown in *Figure 3*, we hypothesized that within-assembly learned connections should reflect the stimulus occurrence probabilities and hence the activation probabilities of the corresponding cell assemblies during spontaneous activity. Therefore, we calculated the total strengths of incoming recurrent synapses on each neuron within the individual cell assemblies (*Figure 4b*). While the distributions of incoming synaptic strengths are similar between cell assemblies coding stimulus 1 and stimulus 2 in the 50-vs-50 case, they look different in the 30-vs-70 case (*Figure 4c*).

Since incoming weights increased more significantly in the cell assembly activated by a more frequent stimulus (i.e., the assembly encoding stimulus 2 in the 30-vs-70 case), we expect that the degree of positive shifts in incoming weight distributions will reflect stimulus probabilities. To examine whether this is indeed the case, we computed the sum of total excitatory incoming weights (i.e., the sum of positive elements of M) over neurons belonging to each assembly after training. We then normalized these excitatory incoming weights over the two assemblies. Interestingly, we found that the normalized excitatory incoming weights for the two assemblies well approximate the empirical probabilities of the two stimuli in both the 50-vs-50 and 30-vs-70 cases (*Figure 4d*). These analyses revealed that recurrent connections learned within assemblies encode biases in probabilistic sensory experiences. Indeed, the elimination of between-assembly excitatory connections did not significantly affect the replay probabilities, as the sampling is driven by strong within-assembly recurrent inputs after learning (*Figure 4—figure supplement 1*).

## Roles of inhibitory plasticity for stabilizing cell assemblies

Experimental and computational results suggest that inhibitory synapses are more robust to spontaneous activity than excitatory synapses and are crucial for maintaining cortical circuit function (*Mongillo et al., 2018*). To see the crucial role of the inhibitory plasticity of G for cell-assembly formation, we compared the spontaneously driven activities in the learned network between two cases, plastic inhibitory connection G versus fixed G, in the 30-vs-70 case. The results show that only a single, highly active assembly self-organizes for fixed inhibitory synapses (*Figure 4—figure supplement 2a*). In contrast, such unstable dynamics do not emerge from plastic inhibitory synapses (*Figure 4—figure supplement 2b*), suggesting the crucial role of inhibitory plasticity in stabilizing spontaneous activity.

To further clarify the functional role of inhibitory plasticity in regulating spontaneous activity, we compared how the self-organized assembly structure of recurrent connections $M$ evolves in the two simulation settings shown in *Figure 4—figure supplement 3a*. In the control model, we turned off the plasticity of $G$ for a while after the cessation of external stimuli but again switched it on, as was previously in *Figure 2*. The cell-assembly structure initially dissipated but eventually reached a well-defined equilibrium structure (*Figure 4—figure supplement 3b*, magenta). Consistent with this, the postsynaptic potentials mediated by connections $M$ and $G$ predicted the normalized firing rate of a postsynaptic excitatory neuron in the control model (*Figure 4—figure supplement 3c*). In striking contrast, the cell-assembly structure rapidly dissipated in the truncated model in which the G-plasticity was kept turned off after the cessation of external stimuli (*Figure 4—figure supplement 3b*, blue). Accordingly, the postsynaptic potentials induced by $M$ and $G$, so was the normalized firing rate, evolved into trivial solutions and almost vanished in the truncated model (*Figure 4—figure supplement 3d*). Only the control model, but not the truncated model, could maintain prediction errors small and nearly constant after the termination of the stimuli (*Figure 4—figure supplement 3e*). These results indicate that maintaining the learned representations requires the continuous tuning of within-assembly inhibition.

## The role of homeostatic regulation of neural activities

As indicated by the weak couplings between cell assemblies, the present mechanism of probability learning differs from the conventional sequence learning mechanisms. Consistent with this, the network trained repetitively by a fixed sequence of patterned inputs does not exhibit stereotyped sequential transitions among cell assemblies (due to the lack of strong inter-assembly excitatory connections; *Figure 4—figure supplement 4*). Indeed, the probability-encoding spontaneous activity emerges in the present model mainly from the within-assembly dynamics driven by strong within-assembly reverberating synaptic input. However, homeostatic variable $h$ also plays a role in maintaining a stable spontaneous network activity after learning (see *Figure 2d*; activity pattern from 5 to 10 s). This is

achieved by the time evolution of $h$, which maintains the firing rate of each neuron in a suitable range by adjusting the threshold and gain of the somatic sigmoidal response function (**Figure 1b**).

Therefore, we explored the role of the homeostatic variable in learning an accurate internal model of the sensory environment. In each neuron, the variable $h$ is updated whenever the membrane potential undergoes an abrupt increase (**Equation 6**). Therefore, the time evolution of $h$ monitors the activity of each neuron over the timescale of seconds, which in turn regulates the neural activity by controlling the activation function (**Figure 4—figure supplement 5a**; **Equations 4 and 5**). When the instantaneous value of $h$ is high, the neuron's excitability is lowered (namely, the gain and threshold of the response function are decreased or increased, respectively: see **Equations 6–8**). This activity regulation is crucial to avoid the trivial solution of the plasticity rules (**Asabuki and Fukai, 2020**) but not critical for sampling with appropriate probabilities. Actually, a model with a fixed value of $h$ still showed spontaneous replay, although the true probability distribution was estimated less accurately (**Figure 4—figure supplement 5b**: **Figure 3f**).

## Learning conditioned prior distributions

The predictive coding hypothesizes that top–down input from higher cortical areas provides prior knowledge about computations in lower cortical areas. This implies in the brain's hierarchical computation that the top–down input conditions the prior distributions in local cortical areas to those relevant to the given context. The proposed learning rules can account for how a conditioned input from other cortical areas conditions the prior distribution in a local cortical circuit.

The neural network consists of two mutually interacting non-overlapping subnetworks of equal sizes, where the subnetworks may represent different cortical areas (**Figure 4—figure supplement 6a**). Subnetwork A was randomly exposed to stimuli 1 and 2 (S1 and S2) with equal probabilities 1/2, whereas subnetwork B was to stimuli 3 and 4 (S3 and S4) with the conditional probabilities 1/3 and 2/3 if S1 was presented to subnetwork A and the conditional probabilities 2/3 and 1/3 if S2 was presented to subnetwork A. After learning, the network model self-organized four cell assemblies, each of which responded preferentially to one of the four stimuli (**Figure 4—figure supplement 6b**). Consistent with this, the self-organized connection matrix represented strong within-assembly connections within each cell assembly and weak between-assembly connections (**Figure 4—figure supplement 6c**). Note that between-assembly connections were inhibitory between assemblies encoding mutually exclusive stimuli, i.e., S1 and S2 and S3 and S4, as they should be. Now, we turned off S3 and S4 to subnetwork B and only applied S1 or S2 to subnetwork A, each at one time. Applying the same stimulus (i.e., S1 or S2) to subnetwork A activated either S3- or S4-coding cell assembly in subnetwork B in a probabilistic manner (**Figure 4—figure supplement 6d**). The cell assemblies evoked in subnetwork B by S1 or S2 to subnetwork A varied the total firing rates approximately in proportion to the conditional probabilities (e.g., P(S3|S1) = 1/3 vs. P(S4|S1) = 2/3) used during learning (**Figure 4—figure supplement 6e**). Note that S3- and S4-coding cell assemblies could become simultaneously active to represent the desired activation probabilities (e.g., a vertical arrow in **Figure 4—figure supplement 6d**). Together, these results indicate that our network can learn prior distributions conditioned by additional inputs through different pathways.

## Replication of biased perceptual decision making in monkeys

Prior knowledge about the environment often biases our percept of the external world. For instance, if we know that two possible stimuli exist and that stimulus A appears more often than stimulus B, we tend to feel that a given stimulus is more likely to be stimulus A than stimulus B. Previously, a similar bias was quantitatively studied in monkeys performing a perceptual decision-making task (**Hanks et al., 2011**). In the experiment, monkeys had to judge the direction (right or left) of the coherent motion of moving dots on a display. When both directions of coherent motion appeared randomly during learning, the monkey showed unbiased choice behaviors. However, if the frequencies of the two motion directions were different, the monkey's choice was biased toward the direction of a more frequent motion stimulus.

We constructed a network model shown in **Figure 5a** to examine whether the present mechanism of spontaneous replay could account for the behavioral bias. The model comprises a recurrent network similar to that used in **Figure 2** and two input neuron groups, L and R, encoding leftward or rightward coherent dot movements, respectively. We modulated the firing rates of these input

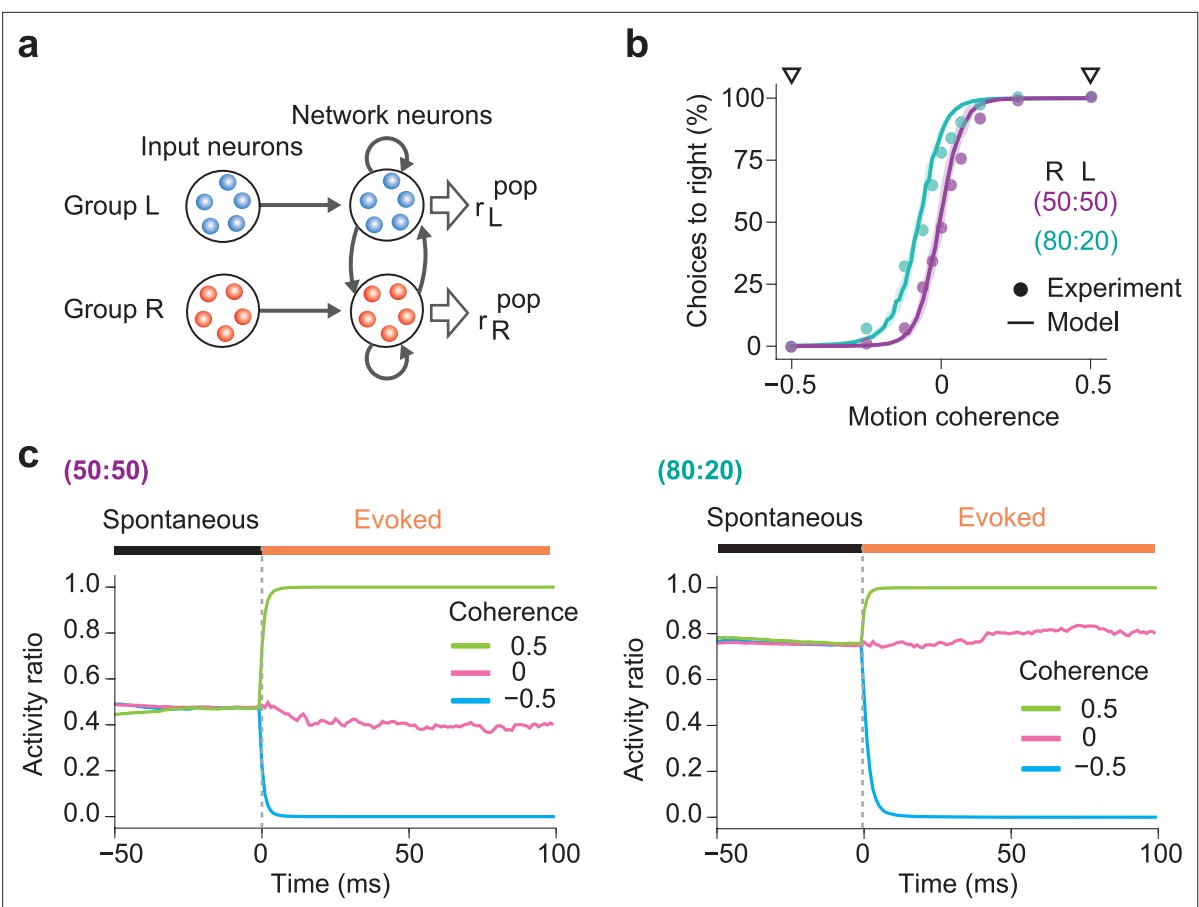

**Figure 5.** Simulations of biased perception of visual motion coherence. (**a**) The network model simulated perceptual decision-making of coherence in random dot motion patterns. In the network shown here, network neurons have already learned two assemblies encoding leftward or rightward movements from input neuron groups L and R. The firing rates of input neuron groups were modulated according to the coherence level Coh of random dot motion patterns (Materials and Methods). (**b**) The choice probabilities of monkeys (circles) and the network model (solid lines) are plotted against the motion coherence in two learning protocols with different prior probabilities. The experimental data were taken from **Hanks et al., 2011**. In the 50:50 protocol, moving dots in the "R" (Coh = 0.5) and "L" (Coh = -0.5) directions were presented randomly with equal probabilities, while in the 80:20 protocol, the "R" and "L" directions were trained with 80% and 20% probabilities, respectively. Shaded areas represent SDs over 20 independent simulations. The computational and experimental results show surprising coincidence without curve fitting. (**c**) Spontaneous and evoked activities of the trained networks are shown for the 50:50 (left) and 80:20 (right) protocols. Evoked responses were calculated for three levels of coherence: Coh = -50%, 0%, and 50%. In both protocols, the activity ratio in spontaneous activity matches the prior probability and gives the baseline for evoked responses. In the 80:20 protocol, the biased priors of "R" and "L" motion stimuli shift the activity ratio in spontaneous activity to an "R"-dominant regime.

neurons in proportion to the coherence of moving dots (Materials and methods). During learning, we trained this model with external stimuli having input coherence Coh of either –0.5 or +0.5 (Materials and methods), where all dots move leftward in the former or rightward in the latter. In so doing, we mimicked the two protocols used in the behavioral experiment of monkeys: in the 50:50 protocol, two stimuli with Coh = ±0.5 were presented randomly with equal probabilities, while in the 80:20 protocol, stimuli with Coh = +0.5 and –0.5 were delivered with probabilities of 80% and 20%, respectively. In the 80:20 protocol, stimuli were highly biased toward a coherent rightward motion.

The network model could explain the biased choices of monkeys surprisingly well. In either training protocol, the recurrent network self-organized two cell assemblies responding selectively to one of the R and L input neuron groups. Then, we examined whether the responses of the self-organized network are consistent with experimental observations by stimulating it with external inputs having various degrees of input coherence. The resultant psychometric curves almost perfectly coincide with those obtained in the experiment (**Figure 5b**). We note that the psychometric curves of the model do not significantly depend on the specific choices of parameter values as far as the network learned

stable spontaneous activity. We did not perform any curve fitting to experimental data, implying that the psychometric curves are free from parameter finetuning.

Biases in the psychometric curves emerged from biased firing rates of spontaneous activity of the self-organized cell assemblies. To show this, we investigated how the activities of the two self-organized cell assemblies change before and after the onset of test stimuli in three relatively simple cases, i.e., Coh = –0.5, 0, and +0.5. *Figure 5c* shows the activity ratio AR between the R-encoding cell assembly and the entire network (Materials and methods) in pre-stimulus spontaneous and post-stimulus-evoked activity. When the network was trained in a non-biased fashion (i.e., in the 50:50 protocol), the activity ratio was close to 0.5 in spontaneous activity, implying that the two cell assemblies had similar activity levels. In contrast, when the network was trained in a biased fashion (i.e., in the 80:20 protocol), the activity ratio in spontaneous activity was close to 0.8, implying that the total spontaneous firing rate of R-encoding cell assembly was four times higher than that of L-encoding cell assembly. Our results show that the spontaneous activity generated by the proposed mechanism can account for the precise relationship between motion coherence and perceptual biases in decision making by monkeys.

## Crucial roles of distinct inhibitory pathways

The model presented so far lacked biological plausibility in several key aspects. Specifically, we assumed that the recurrent connections $M$ could change its sign through plasticity and be either excitatory or inhibitory, while the inhibitory connection $G$ was restricted to being inhibitory only. This setting does not reflect the biological constraint that synapses maintain a consistent excitatory or inhibitory type. Furthermore, due to this unconstrained recurrent connectivity $M$, the original model had two types of inhibitory connections (i.e., the negative part of $M$ and the inhibitory connection $G$) without providing a clear computational role for each type of inhibition.

To address these limitations and to understand the role of the two types of inhibition, we considered a novel architecture in which all recurrent connections are constrained to be either exclusively excitatory or inhibitory, maintaining their sign throughout the learning process. The refined model includes two different types of inhibitory connections (i.e., $M_{inh}$ and $G$), each serving a specific computational purpose: minimizing prediction error and maintaining the excitatory–inhibitory balance. In combination with the excitatory connection $M_{exc}$, the $M_{inh}$ connections are trained to minimize the prediction error between somatic and dendritic activity, as considered in the original M connection in *Figure 1*. We found that the trained $M_{inh}$ connections introduce competition among cell-assembly activities by forming strong connections between assemblies (*Figure 6b*), allowing the network to effectively sample and replay the activities of individual assemblies. In contrast, inhibitory connections $G$ were trained to balance network dynamics, as in

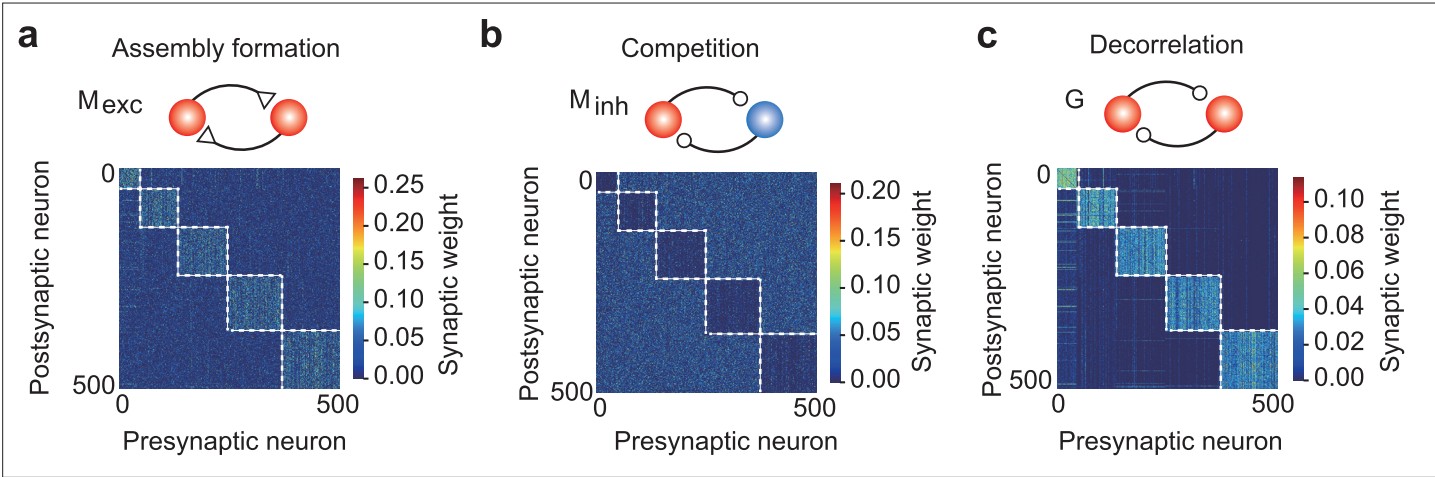

**Figure 6.** A network model with distinct excitatory and inhibitory connections. (a) Strong excitatory connections were formed within assemblies. (b) The first type of recurrent inhibitory connections, $M_{inh}$, became stronger between assemblies, enhancing assembly competition. (c) The second type of inhibitory connections G were strengthened within assemblies to balance the strong excitatory inputs.

the original setting. We found that the inhibitory $G$ connections form strong intra-assembly inhibition (*Figure 6c*), which balances the strong excitatory connections that arise within cell assemblies through plasticity (*Figure 6a*).

In summary, the dual inhibitory mechanism allows the network to perform the reactivation of different cell assemblies while regulating their internal dynamics. The prediction-error-minimizing inhibitory connections $M_{inh}$ facilitate selecting and activating specific assemblies through competition such that the learned probabilities are replayed. In contrast, the network-balancing inhibitory connections $G$ prevent runaway excitation within active assemblies.

## An elaborate network model with distinct excitatory and inhibitory neuron pools

The predictive learning rule performed well in training the nDL model to learn the probabilistic structure of the stimulus-evoked activity patterns. However, whether the same learning rule works in a more realistic neural network is yet to be investigated. To examine this, we constructed an elaborate network model (DL model) consisting of distinct excitatory and inhibitory neuron pools, obeying Dale's law (*Figure 7a*). The nDL model suggested the essential roles of inhibitory plasticity in maintaining excitation–inhibition balance and generating an appropriate number of cell assemblies. To achieve these functions, inhibitory neurons in the DL model project to excitatory and other inhibitory neurons via two synaptic paths (*Figure 7b*), motivated by the results shown in *Figure 6*. In path 1, inhibitory connections alone predict the postsynaptic activity, whereas inhibitory and excitatory connections jointly predict the activity of the postsynaptic neuron in path 2 (Materials and methods). All synapses in the DL model are subject to the predictive learning rule. We trained the DL model with three input neuron groups while varying their activation probabilities. As in the nDL model, the DL model self-organized three cell assemblies activated selectively by the three input neuron groups (*Figure 7—figure supplement 1a*). Furthermore, in the absence of external stimuli, the DL model spontaneously replayed these assemblies with the assembly activity ratios in proportion to the occurrence probabilities of the corresponding stimuli during learning (*Figure 7c*).

The two inhibitory paths divided their labors consistent with the results shown in *Figure 6*. To see this, we investigated the connectivity structures learned by these paths. In path 1, inhibitory connections were primarily found on excitatory neurons in the same assemblies (*Figure 7d*, top). In contrast, in path 2, inhibitory connections were stronger on excitatory neurons in different assemblies than those in the same assemblies (*Figure 7d*, bottom). On both excitatory and inhibitory neurons, the total inhibition (i.e., path 1 + path 2) was balanced with excitation (*Figure 7e*). *Figure 7f* summarizes the connectivity structure of the DL model. Excitatory neurons in a cell-assembly project to inhibitory neurons in the same assembly. Then, these inhibitory neurons project back to excitatory neurons in the same or different assemblies via paths 1 and 2. Interestingly, lateral inhibition through path 1 is more potent between excitatory neurons within each cell assembly than between different assemblies (*Figure 7g*). In contrast, path 2 mediates equally strong within- and between-assembly inhibition.

We can understand the necessity of the two inhibitory paths based on the dynamical properties of competitive neural networks. *Figure 7h* displays the effective competitive network of excitatory cell assemblies suggested by the above results. Both paths 1 and 2 contribute to within-assembly inhibition among excitatory neurons, whereas between-assembly inhibition (i.e., lateral inhibition) mainly comes from path 2. In a competitive network, the lateral inhibition to self-inhibition strength ratio determines the number of winners having non-vanishing activities: the higher the ratio is, the smaller the number of winners is (*Fukai and Tanaka, 1997*). Therefore, self-organizing the same number of excitatory cell assemblies as that of external stimuli requires tuning the balance between the within- and between-assembly inhibitions. This tuning during learning is likely easier when the network has two independently learnable inhibitory circuits. Indeed, a network model with only one inhibitory path rarely succeeded in encoding and replaying all stimuli used in learning (*Figure 7—figure supplement 1b, c*).

In summary, we have shown the roles of distinct recurrent inhibitory connections. Using a network consisting of excitatory and inhibitory populations, we have shown that distinct inhibitory circuits are necessary to generate within- and between-assembly competition crucial to maintain the stability of learned multiple assemblies.

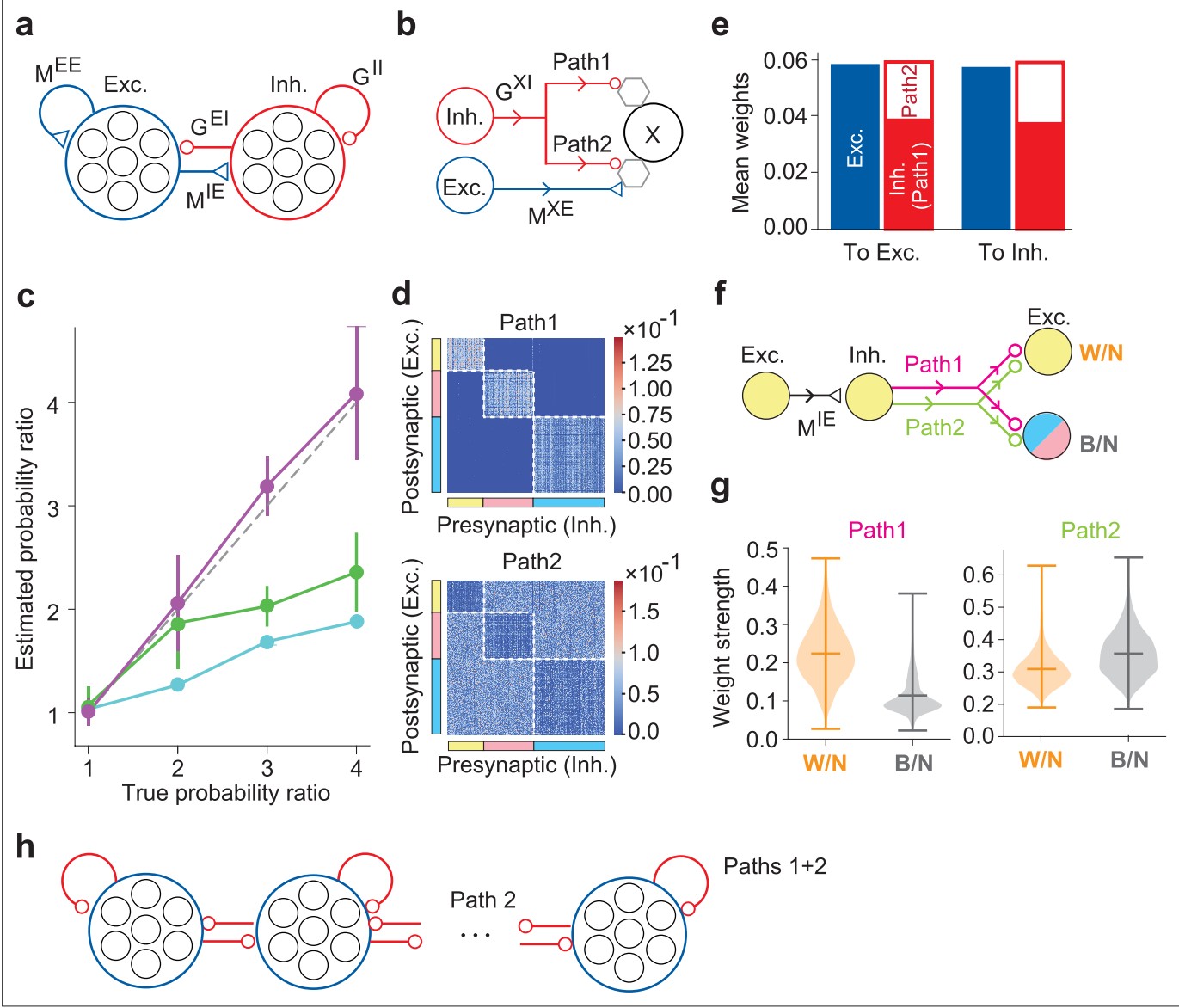

**Figure 7.** The DL model of excitatory and inhibitory cell assemblies. (a) This model consists of distinct excitatory and inhibitory neuron pools, obeying Dale's law. (b) Each inhibitory neuron projects to another neuron X through two inhibitory paths, path 1 and path 2, where the index X refers to an excitatory or an inhibitory postsynaptic neuron. Hexagons represent minimal units for prediction and learning in the neuron model and may correspond to dendrites, which were not modeled explicitly. (c) The probability ratios estimated by numerical simulations are plotted for the assembly activity ratios (purple), firing rate ratios (cyan), and assembly size ratios (green) as functions of the true probability ratio of external stimuli. Error bars indicate SEs calculated over five simulation trials with different initial states of neurons and synaptic weights in each parameter setting. (d) Inhibitory connection matrices are shown for path 1 and path 2. (e) The mean weights of self-organized synapses on excitatory and inhibitory postsynaptic neurons are shown. (f) Within-assembly and between-assembly connectivity patterns of excitatory and inhibitory neurons are shown. Colors indicate three cell assemblies self-organized. (g) The strengths of lateral inhibitions within-(W/N) and between-assemblies (B/N) are shown for paths 1 and 2. Horizontal bars show the medians and quartiles. (h) The resultant connectivity pattern suggests an effective competitive network between excitatory assemblies with self-(within-assembly) and lateral (between-assembly) inhibition.

The online version of this article includes the following figure supplement(s) for figure 7:

**Figure supplement 1.** The coexistence of the two inhibitory paths is crucial for learning.

## Discussion

Having proper generative models is crucial for accurately predicting statistical events. The brain is thought to improve the prediction accuracy of inference by learning internal generative models of the environment. These models are presumably generated through multiple mechanisms. For instance,

the predictive coding hypothesizes that top–down cortical inputs provide lower sensory areas with prior information about sensory experiences (*Friston, 2010*; *Bastos et al., 2012*; *Keller and Mrsic-Flogel, 2018*). However, experimental evidence also suggests that spontaneous activity represents an optimal model of the environment in sensory cortices. This study proposed a biologically plausible mechanism to learn such a model, or priors for experiences, with the brain's internal dynamics.

Our model adopted a single predictive learning principle for the plasticity of excitatory and inhibitory synapses to learn the replay of probabilistic experiences. On each neuron, excitatory and inhibitory synaptic weights undergo plastic changes to improve their independent predictions on the cell's firing. This was done by minimizing the mismatch between the output firing rate and the network predictions (*Equations 9 and 17*). This simple learning rule showed excellent performance in a simplified network model and in a more realistic model obeying Dale's law. The latter model predicts a division of labor between two inhibitory paths. Intriguingly, the inhibitory path 2 of this model resembles interpyramidal inhibitory connections driven directly by nearby pyramidal cells (*Ren et al., 2007*). In both models, inhibitory synaptic plasticity plays a crucial role in learning an accurate internal model by maintaining excitation–inhibition balance and decorrelating cell-assembly activities (*Vogels et al., 2013*; *Sprekeler, 2017*). It should be noted that while several network models that perform error-based computations like ours exploit only inhibitory recurrent plasticity (*Mikulasch et al., 2021*; *Mackwood et al., 2021*; *Hertäg and Clopath, 2022*; *Mikulasch et al., 2023*), our model learns to reproduce appropriate statistics by modifying both excitatory and inhibitory recurrent connections.

Various models have been proposed to account for neural mechanisms of Bayesian computation by the brain (*Tully et al., 2014*; *Kappel et al., 2015*; *Hiratani and Fukai, 2018*; *Hiratani and Latham, 2020*; *Aitchison et al., 2021*; *Ma et al., 2006*; *Deneve, 2008*; *Nessler et al., 2013*; *Hiratani and Fukai, 2016*; *Huang and Rao, 2016*; *Isomura et al., 2022*; *Friston, 2010*; *Bastos et al., 2012*; *Keller and Mrsic-Flogel, 2018*). Typically, these models embed prior knowledge on sensory experiences into the wiring patterns of afferent (and sometimes also recurrent) synaptic inputs such that these inputs can evoke the learned activity patterns associated with the prior knowledge. The present model differs from the previous models in several aspects: (1) The model segments repeated stimuli to be remembered in an unsupervised fashion; (2) Then it generates cell assemblies encoding the segmented stimuli; (3) Finally, it replays these cell assemblies spontaneously with learned probabilities. Note that the same learning rules enable the network to perform all necessary computations for (1)–(3). To our knowledge, our model is the first to perform all these steps for encoding an optimal model of the environment into spontaneous network activity.

The present mechanism of memory formation differs from the previous ones that self-organize cell assemblies through Hebbian learning rules (*Vogels et al., 2011*; *Hiratani and Fukai, 2014*; *Litwin-Kumar and Doiron, 2014*; *Zenke et al., 2015*; *Triplett et al., 2018*; *Montangie et al., 2020*). First, these mechanisms did not aim for explicit statistical modeling of the environment. Second, the previous studies suggested that the orchestration of multiple plasticity rules, including inhibitory plasticity and homeostatic synaptic scaling, enables the maintenance of cell assemblies (however, see *Manz and Memmesheimer, 2023*). For instance, in STDP, slight changes in the relative times of pre- and postsynaptic spikes can change the polarity of synaptic modifications, implying that STDP requires a mechanism to keep synaptic weights finite (*Kempter et al., 1999*; *Song et al., 2000*; *Masquelier et al., 2008*). In contrast, our learning rule, which induces either long-term potentiation or depression according to the sign of the prediction error calculated independently within each postsynaptic neuron, does not suffer such instability.

Our model predicts a novel intracellular process that regulates the neuron's dynamic range according to the history of its subthreshold dynamics. This process plays two important roles in the statistical modeling of our model. First, it avoids the trivial solution (i.e., the zero-weight solution) of our unsupervised predictive learning by homeostatically regulating neurons' intrinsic excitability. Second, the intracellular process cooperates with reverberating synaptic inputs within each cell assembly to generate spontaneous replay activity. We have shown that intracellular homeostasis enhances the sampling from learned distribution without relying on the recurrences among assemblies. This mechanism contrasts with the previous sampling-based models that rely on the transition dynamics between cell assemblies (*Buesing et al., 2011*; *Bill et al., 2015*). How neural systems implement the proposed homeostasis is an open question.

Previous computational models have demonstrated that recurrent networks with Hebbian-like plasticity can learn appropriate Markovian statistics (*Kappel et al., 2015*; *Asabuki and Clopath, 2024*). However, our model differs conceptually from these previous models. Kappel et al. showed that STDP in winner-take-all circuits can approximate online learning of hidden Markov models. A key difference with our model is that their neural representations acquire sequences using Markovian sampling dynamics, whereas our model does not depend on such ordered sampling. Specifically, in their model, sequential sampling arises from the off-diagonal elements learned in the recurrent connections (i.e., between-assembly connections). In contrast, our network learns to generate a stochastic reactivation of cell assemblies solely by within-assembly connections. A similar argument can be made for the Asabuki and Clopath paper as well. Further, while our model introduced plasticity at all types of synaptic connections, the previous model assumed plasticity only at recurrent synapses projecting onto the excitatory neurons. In addition, unlike our model, the cell-assembly memberships need to be preconfigured in the previous model.

The proposed mechanism can account for the behavioral biases observed in perceptual decision making (*Hanks et al., 2011*). This behavioral experiment quantitatively clarified how the difference in the probability between sensory experiences during learning biases the alternative choice behavior of monkeys. In our model, two cell assemblies encoding the different stimuli are replayed at the total firing rates proportional to the corresponding occurrence probabilities. Our results suggest that the difference in spontaneous firing rates of cell assemblies is sufficient to explain the behavioral biases of monkeys. However, other mechanisms, such as biased top–down input, cannot be excluded.

What could be the advantages of coding prior distributions into spontaneous activity over other ways of probability coding? First, spontaneous replay activities in lower cortical areas may provide training data for modeling by higher cortical areas, promoting hierarchical statistical modeling in predictive coding. This is analogous to the situation where hippocampal engram cells are replayed to reinforce the activity patterns of cortical engrams for memory consolidation during sleep (*Tonegawa et al., 2018*; *Ghandour et al., 2019*; *Klinzing et al., 2019*; *Takehara-Nishiuchi, 2021*). Memory reinforcement by activity replay has also been studied in machine intelligence (*Dayan et al., 1995*; *Goodfellow, 2014*; *Luczak et al., 2022*). Second, spontaneous replay of internal models may support knowledge generalization during sleep. It was recently reported that a transitive inference task requires post-learning sleep (*Abdou et al., 2021*). In this task, mice had to infer a correct reward delivery rule in a novel behavioral situation from the outcomes of past experiences. The mice failed to generalize the learned rules if the activity of the anterior cingulate cortex was suppressed during post-learning sleep, suggesting that dynamic interactions among rule-coding cortical neurons in spontaneous activity are crucial for rule generalization. Clarifying how spontaneous brain activity generalizes the learned internal models is an intriguing open question.

## Methods
### Neural network model

Below, we first describe the model architecture and learning rule for the nDL model (i.e., single population violating Dale's law). Details of the simulation of distinct excitatory and inhibitory populations will be explained later. Unless otherwise stated, recurrent neural networks used in this study consist of $N (= 500)$ Poisson neurons, which generate spikes according to a non-stationary Poisson process with rate $\varphi(u)$, where $\varphi(\cdot)$ is a dynamics sigmoidal function, which we will explain later. The membrane potential $u$ of neuron $i$ at time $t$ is given as follows:

$$u_i(t) = \sum_{k=1}^{K} W_{ik} x_k(t) + \sum_{k=1}^{N} (M_{ik} - G_{ik}) y_k(t),$$ (1)

where $K$ is the number of input neurons. In some simulations, the network model had more than one input neuron group, although the number of input neuron groups is not explicitly shown in *Equation 1*. Three matrices $W \in \mathbb{R}^{N \times K}$, $M \in \mathbb{R}^{N \times N}$, and $G \in \mathbb{R}^{N \times N}$ represent the weights of afferent synaptic connections, recurrent synaptic connections, and inhibitory-only connections, respectively, on neurons in the recurrent network. These synaptic connections are all-to-all. In terms of the kernel function

$$\varepsilon(s) = \exp(-s/\tau) \cdot \Theta(s),$$ (2)

recurrent input and afferent input to neuron $i$ are calculated as

$$x_i(t) = \sum_{t' \in t_{\text{aff}}^f} \varepsilon\left(t - t'\right),$$  (3a)

$$y_i(t) = \sum_{t' \in t_{\text{rec}}^f} \varepsilon\left(t - t'\right),$$  (3b)

where $\tau$ stands for the membrane time constant, $t_{\text{aff}}^f$ and $t_{\text{rec}}^f$ for the time sets of afferent and recurrent presynaptic spikes, and $\Theta(\cdot)$ for the Heaviside function. Throughout this study, $\tau = 15$ ms.

The instantaneous firing rate $f_i(t)$ of each neuron is given as

$$f_i(t) = \hat{\varphi}\left(u_i(t); h_i\right),$$  (4)

in terms of a dynamical sigmoidal response function $\hat{\varphi}$:

$$\hat{\varphi}(u_i; h_i) = \varphi_0 \left[1 + \exp\left[g\beta(h_i)\left(-u_i + g\theta(h_i)\right)\right]\right]^{-1},$$  (5)

with a constant value of $g = 3$ and we have dropped the explicit time dependence in our notation for the sake of simplicity. Here, the dynamical variable $h$ is determined by the history of the membrane potential:

$$\begin{aligned} \dot{h}_i &= -\tau_h^{-1} h_i, \quad \text{if } h_i > u_i, \\ h_i &\leftarrow u_i, \qquad \text{otherwise.} \end{aligned}$$  (6)

The maximum instantaneous firing rate $\varphi_0$ is 50 Hz and $\tau_h = 10$ s. Through **Equation 6**, $h_i$ tracks the maximum value of the membrane potential $u_i$ in a time window of approximately the length $\tau_h$ in the immediate past. The value of $h$ is utilized to regulate the gain $\beta$ and threshold $\theta$ of the sigmoidal response function as follows:

$$\beta(h_i) = h_i^{-1}\beta_0,$$  (7)

$$\theta(h_i) = h_i\theta_0,$$  (8)

where the values of constant parameters are $\beta_0 = 5$, and $\theta_0 = 1$. Neuron $i$ generates a Poisson spike train at the instantaneous firing rate of $f_i(t)$. While a small value of h leads to a steep slope of our activation function (**Equation 7**), we have shown numerically that this does not lead to a problem in neural dynamics. Further, the saturation part of the sigmoidal function is crucial for stable formation of assemblies.

## Learning rules

We first explain the plasticity rule for feedforward connections. Synaptic connections were modified to minimize the Kullback–Leibler divergence (KL-divergence) between two Poisson distributions associated with the neuron's output and the feedforward activity over a sufficiently long period $T$:

$$\mathcal{L}_W = \int_0^T dt \sum_{i=1}^N D_{KL}\left[f_i(t) \| \varphi\left(v_i^W(t)\right)\right],$$  (9)

where $v_i^W$ is a feedforward prediction of a firing rate, defined as:

$$v_i^W = \sum_{j=1}^K W_{ij} \cdot x_j,$$  (10)

and $f_i$ is the firing rate of $i$th neuron. The function $\varphi$ is a static sigmoidal function, defined as

$$\varphi(v_i) = \varphi_0 \left[1 + \exp\left[g\beta_0\left(-v_i + g\theta_0\right)\right]\right]^{-1}.$$  (11)

The above cost function evaluates to what extent the feedforward potential predicts the activity of postsynaptic neurons (**Asabuki and Fukai, 2020**). We have previously shown that taking the gradient of the cost function in **Equation 9** derives the online plasticity rule for the feedforward connections as

$$\Delta W_{ij} = \eta \varphi_0^{-1} \left( 1 - \frac{\varphi\left(v_i^W\right)}{\varphi_0} \right) \left[ f_i - \varphi\left(v_i^W\right) \right],$$

(12)

where $\epsilon$ is a learning rate and was set to $\epsilon = 10^{-4}$, unless otherwise specified. Here, we have dropped the explicit time dependence in our notation for the sake of simplicity.

Similarly, the recurrent connections were modified to minimize the following cost function:

$$\mathcal{L}_M = \int_0^T dt \sum_{i=1}^N D_{KL} \left[ f_i(t) \,\|\, \varphi\left(v_i^M(t)\right) \right],$$

(13)

where $v_i^M = \sum_{j=1}^N M_{ij} \cdot y_j$ is a recurrent prediction. Similar to the feedforward plasticity, the gradient descent of the above cost function leads to the following plasticity rule:

$$\Delta M_{ij} = \eta \varphi_0^{-1} \left( 1 - \frac{\varphi\left(v_i^M\right)}{\varphi_0} \right) \left[ f_i - \varphi\left(v_i^M\right) \right].$$

(14)

The derived recurrent plasticity rule suggests that the recurrent prediction learns the statistical model of the evoked activity, which in turn allows the network to replay the learned internal model.

In addition to the above plasticity rules, we defined the cost function for the inhibitory plasticity as

$$\mathcal{L}_G = \sum_{i=1}^N \left[ f_i(t) - \varphi\left(v_i^G(t)\right) \right]^2,$$

(15)

where $v_i^G$ is the inhibitory input onto postsynaptic neuron via inhibitory connection $G$:

$$v_i^G = \sum_{j=1}^N G_{ij} \cdot y_j.$$

(16)

Again, by taking the gradient of $\mathcal{L}_G$ with respect to $G_{ij}$ derive the following inhibitory plasticity rule to keep the network dynamics balanced:

$$\begin{aligned}
\Delta G_{ij} &\propto -\frac{\partial \mathcal{L}_G}{\partial G_{ij}} \\
&\propto \left[ f_i - \varphi\left(v_i^G\right) \right] \times \frac{\partial \varphi\left(v_i^G\right)}{\partial G_{ij}} \\
&\propto \varphi\left(v_i^G\right) \left( 1 - \frac{\varphi\left(v_i^G\right)}{\varphi_0} \right) \cdot \left[ f_i - \varphi\left(v_i^G\right) \right] \cdot y_j.
\end{aligned}$$

(17)

While the resultant rule is not the same as feedforward and recurrent plasticity rules, all of these rules are similar in a sense that the weight updates are proportional to the prediction error and the presynaptic activity. We therefore assumed the following rule for the inhibitory plasticity, which has the same structure as the rest of the plasticity rules that we have already explained:

$$\Delta G_{ij} = \eta \varphi_0^{-1} \left( 1 - \frac{\varphi\left(v_i^G\right)}{\varphi_0} \right) \left[ f_i - \varphi\left(v_i^G\right) \right].$$

(18)

We have shown by numerical simulation that the rule keeps the network dynamics balanced.

Initial values of $W$ and $M$ are sampled from Gaussian distributions with the mean 0 and variances $0.1/\sqrt{K}$ and $0.1/\sqrt{N}$, respectively. During learning, the elements of $W$ and $M$ can take both positive and negative values. After sufficient learning, the postsynaptic potentials $v_i^W$ and $v_i^M$ on neuron on

neuron $i$ converge to a common value of $v_i$. Therefore, $\varphi\left(v_i^W\right) \approx \varphi\left(v_i^M\right) \approx \varphi\left(v_i\right) \approx f_i$, implying that the postsynaptic potentials of afferent and recurrent synaptic inputs to neuron $i$ can both predict its output $f_i$ after learning. The initial values of $G$ are uniformly set to $1/\sqrt{N}$, and its elements are truncated to non-negative values during learning. This implies that $v_i^G$ does not become negative. After learning, $\varphi\left(v_i^G\right) \approx f_i$ is satisfied. Although some elements of $M$ may give recurrent inhibitory connections, modifiable connections in $G$ are necessary to encode all external inputs into specific cell assemblies.

## Stimulation protocols

Feedforward input to the recurrent network consisted of $K$ Poisson spike trains with a background firing rate of 2 Hz. The input randomly presented $n$ non-overlapping patterns of 100 spike trains (the duration 100 ms and the mean frequency 50 Hz), one at a time, with pattern-to-pattern intervals of 100 ms. Therefore, the number of input neurons and patterns satisfies the relationship of $K = 100 \times n$. For simplicity, we simulated the constant-interval case, but using irregular intervals does not change the essential results. The value of $n$ varies from task to task, and the values for each figure are as follows: $n = 5$ (*Figure 2c-e*, *Figure 3*, *Figure 6*); $n = 2$ (*Figure 2a-b*, *Figure 4-Figure 5*); $n = 3$ (*Figure 7*). The typical time length required for the convergence of learning is 1000 s.

## Measures for cell-assembly activities

Here, we explain the measures used in *Figure 3*. We calculated the firing rate ratio of cell assembly 1 in *Figure 3c* as follows:

$$\text{Firing rate ratio} = \frac{r_i^{(1)}}{\left(\sum_{j=2}^{5} \frac{1}{N_j} \sum_{i=1}^{N_j} r_i^{(j)}\right)/4}, \tag{19}$$

using the average firing rate $r_i^{(j)}$ of the $i$th neuron in cell assembly $j$ and the number $N_j$ of neurons belonging to the cell assembly. Similarly, we defined the assembly size ratio of cell assembly 1 as

$$\text{Assembly size ratio} = \frac{N_1}{\left(\sum_{j=2}^{5} N_j\right)/4}, \tag{20}$$

in *Figure 3d* and assembly activity ratio of cell assembly 1 as

$$\text{Assembly activity ratio} = \frac{r_{\text{pop}}^{(1)}}{\left(\sum_{i=2}^{5} r_{\text{pop}}^{(i)}\right)/4}, \tag{21}$$

in *Figure 3e*. Here, $r_{\text{pop}}^{(i)}$ represents the population neural activity of cell assembly $i$:

$$r_{\text{pop}}^{(j)} \equiv \sum_{i=1}^{N_j} r_i^{(j)} \tag{22}$$

## Simulations of perceptual decision making

In each learning trial, we trained the network with either leftward or rightward dot movement represented by the corresponding input neurons firing at $r_{\max} = 50\,\text{Hz}$ In test trials, we defined input coherence as $\text{Coh} = \rho_R - 0.5$ according to *Hanks et al., 2011*, where $\rho_R$ is the ratio of R input neurons to the sum of R and L input neurons in firing rate. The value of Coh ranges between –0.5 (all dots moving leftward) and +0.5 (all dots moving rightward). Then, in test trials for input coherence Coh, we generated Poisson spike trains of R and L input neurons at the rates $(\text{Coh} + 0.5)\, r_{\max}$ and $(-\text{Coh} + 0.5)\, r_{\max}$, respectively.

In *Figure 5c*, we calculated the activity ratio (AR) as

$$\mathrm{AR} = \frac{r_{\mathrm{R}}^{\mathrm{pop}}}{r_{\mathrm{R}}^{\mathrm{pop}} + r_{\mathrm{L}}^{\mathrm{pop}}}, \tag{23}$$

where $r_{\mathrm{R}}^{\mathrm{pop}}$ and $r_{\mathrm{L}}^{\mathrm{pop}}$ represent the average population firing rates of R- and L-encoding cell assemblies, respectively. In *Figure 5b*, we defined 'choices to right' as

$$\text{Choices to right} = \mathrm{AR} \times 100 \, (\%) \,. \tag{24}$$

## A network model with distinct excitatory and inhibitory synapses

Here, we explain the network model and the plasticity rules used in *Figure 6*. The network consists of 500 neurons, and the membrane potential of a neuron $i$ at time $t$ is given as follows:

$$u_i(t) = \underbrace{\sum_{k=1}^{K} \mathrm{W}_{ik} x_k(t)}_{=:v_i^{\mathrm{W}}} + \underbrace{\left[\sum_{n=1}^{N} \mathrm{M}_{in}^{\mathrm{exc}} y_n(t)\right]}_{=:v_i^{\mathrm{M}}(\mathrm{exc})} - \underbrace{\left[\sum_{n=1}^{N} \mathrm{M}_{in}^{\mathrm{inh}} y_n(t)\right]}_{=:v_i^{\mathrm{M}}(\mathrm{inh})}$$
$$- \underbrace{\sum_{n=1}^{N} \mathrm{G}_{in} y_n(t)}_{=:v_i^{\mathrm{G}}}, \tag{25}$$

where $\{\mathrm{W}_{ik}\}$ is afferent synaptic weights, which are a mixture of excitatory and inhibitory connections as in the nDL model. The weights of recurrent excitatory synapses are $\{\mathrm{M}_{in}^{\mathrm{exc}}\}$. Here, we considered two types of recurrent inhibitory connections, denoted by $\mathrm{M}^{\mathrm{inh}}$ and $G$, respectively. Here, we assumed that half of the recurrent connections were assumed to be excitatory and the remaining connections were all inhibitory, half of which were $\mathrm{M}^{\mathrm{inh}}$ and the other half were G. We modified these weights according to the following equations:

$$\Delta \mathrm{W}_{ij} = \eta \mathcal{E}(f_i, v_i^{\mathrm{W}}) x_j, \tag{26a}$$

$$\Delta \mathrm{M}_{ij}^{\mathrm{exc}} = \eta \mathcal{E}\left(f_i, v_i^{\mathrm{M}(\mathrm{exc})} - v_i^{\mathrm{M}(\mathrm{inh})}\right) y_j, \tag{26b}$$

$$\Delta \mathrm{M}_{ij}^{\mathrm{exc}} = -\eta \mathcal{E}\left(f_i, v_i^{\mathrm{M}(\mathrm{exc})} - v_i^{\mathrm{M}(\mathrm{inh})}\right) y_j, \tag{26c}$$

$$\Delta G_{ij} = \eta \mathcal{E}\left(f_i, v_i^{G}\right) y_j, \tag{26d}$$

where $\mathcal{E}\left(f_i, v_i\right)$ is the error term defined as

$$\mathcal{E}\left(f_i, v_i\right) \varphi_0^{-1} \left(1 - \frac{\varphi\left(v_i\right)}{\varphi_0}\right) \left[f_i - \varphi\left(v_i\right)\right]. \tag{27}$$

At each time step during learning, we truncated all weights of recurrent connections to non-negative values during learning.

## A network model with distinct excitatory and inhibitory neuron populations

Here, we explain the architecture of the model used in *Figure 7*. The network consists of $N_{\mathrm{E}} \, (= 500)$ excitatory and $N_{\mathrm{I}} \, (= 500)$ inhibitory neurons. The membrane potential of a neuron $i$ of a population X $(=\mathrm{E}$ or I$)$ at time $t$ is given as follows:

$$u_i^{\mathrm{X}}(t) = \underbrace{\sum_{k=1}^{K} \mathrm{W}_{ik}^{\mathrm{X}} x_k(t)}_{=:v_i^{\mathrm{W}}} + \underbrace{\left[\sum_{l=1}^{N_E} \mathrm{M}_{il}^{\mathrm{XE}} y_l^E(t) - \sum_{m=1}^{N_I} \mathrm{G}_{im}^{\mathrm{XI(path2)}} y_m^I(t)\right]}_{=:v_i^{\mathrm{M(2)}}} ,$$

$$\underbrace{- \sum_{m=1}^{N_I} \mathrm{G}_{im}^{\mathrm{XI(path1)}} y_m^I(t)}_{=:v_i^{\mathrm{M(1)}}},$$

(28)

where $\left\{W_{ik}^X\right\}$ is afferent synaptic weights, which are a mixture of excitatory and inhibitory connections as in the nDL model. The weights of recurrent excitatory synapses are $\{\mathrm{M}_{il}^{\mathrm{XE}}\}$. Here, we considered two types of recurrent inhibitory connections (i.e., path 1 and path 2), denoted by $\mathrm{G}_{\mathrm{Im}}^{\mathrm{XI(path1)}}$ and $\mathrm{G}_{\mathrm{Im}}^{\mathrm{XI(path2)}}$, respectively. Using the same definitions of the error term as in *Equation 27*, we modified these weights according to the following equations:

$$\Delta \mathrm{W}_{ij}^{\mathrm{X}} = \eta \mathcal{E}\left(f_i, v_i^{\mathrm{W}}\right) x_j,$$

(29a)

$$\Delta \mathrm{M}_{ij}^{\mathrm{XE}} = \eta \mathcal{E}\left(f_i, v_i^{\mathrm{M(2)}}\right) y_j^E,$$

(29b)

$$\Delta \mathrm{G}_{ij}^{\mathrm{XI(path2)}} = -\eta \mathcal{E}\left(f_i, v_i^{\mathrm{M(2)}}\right) y_j^I,$$

(29c)

$$\Delta \mathrm{G}_{ij}^{\mathrm{XI(path1)}} = \eta \mathcal{E}\left(f_i, v_i^{\mathrm{M(1)}}\right) y_j^I.$$

(29d)

To satisfy Dale's law, we truncated all weights of recurrent connections to non-negative values during learning.

In *Figure 7g*, we measured the lateral inhibition between excitatory neurons via path 1 by calculating:

$$\left[\mathrm{W}_{i,j}^{\mathrm{LI}}\right] = \sum_{k=1}^{N_E} \mathrm{G}_{ik}^{\mathrm{EI(path1)}} \mathrm{M}_{kj}^{\mathrm{IE}}.$$

(30)

Lateral inhibition via path 2 was calculated in a similar fashion.

## Simulation details

All simulations were performed in customized Python3 code written by TA with numpy 1.17.3 and scipy 0.18. Differential equations were numerically integrated using an Euler method with integration time steps of 1 ms.

## Data availability

Code is provided on the GitHub repository (*Asabuki, 2024*). https://github.com/TAsabuki/PriorNet_codes, (copy archived at *Asabuki, 2025*).

## Acknowledgements

The authors express their sincere thanks to Yukiko Goda for her valuable comments on our manuscript. This work was supported by KAKENHI (nos. 18H05213 and 19H04994) to TF.

## Additional information

### Funding

| Funder | Grant reference number | Author |
|---|---|---|
| KAKENHI | 18H05213 | Tomoki Fukai |

| Funder | Grant reference number | Author |
|--------|------------------------|--------|
| KAKENHI | 19H04994 | Tomoki Fukai |

The funders had no role in study design, data collection, and interpretation, or the decision to submit the work for publication.

## Author contributions

Toshitake Asabuki, Conceptualization, Data curation, Software, Formal analysis, Investigation, Visualization, Methodology, Writing – original draft, Writing – review and editing; Tomoki Fukai, Conceptualization, Supervision, Funding acquisition, Writing – original draft, Writing – review and editing

## Author ORCIDs

Toshitake Asabuki ⓘ https://orcid.org/0000-0003-0951-5791
Tomoki Fukai ⓘ http://orcid.org/0000-0001-6977-5638

Reviewer #1 (Public review): https://doi.org/10.7554/eLife.92712.3.sa1
Reviewer #2 (Public review): https://doi.org/10.7554/eLife.92712.3.sa2
Reviewer #3 (Public review): https://doi.org/10.7554/eLife.92712.3.sa3
Author response https://doi.org/10.7554/eLife.92712.3.sa4

---

# Additional files

## Supplementary files

MDAR checklist

## Data availability

The current manuscript is a computational study, so no data have been generated for this manuscript. Modeling code is uploaded as Source code.

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
