## [Editor Report · eLife Assessment]

This **valuable** study investigates how biologically plausible learning mechanisms can support assembly formation that encodes statistics of the environment, by enabling neural sampling that is based on within-assembly connectivity strength. It **convincingly** shows that assembly formation can emerge from predictive plasticity in excitatory synapses, while two types of plasticity in inhibitory synapses are required: inhibitory homeostatic (predictive) plasticity and inhibitory competitive (anti-predictive) plasticity.

---

## [Referee Report · Reviewer #1 (Public review)]

The authors have successfully addressed most of the issues raised in the first review. Nevertheless, some of the mentioned problems require further attention, mostly regarding the formal derivation of the learning rules, as well as connections to previous research.

Regarding the derivations of learning rules: The authors have provided Goal functions for each of the plastic neural connections to give some insight into what these connections do. However, as I understand, this does not address the main concern raised in the previous review: Why do these rules lead to overall network dynamics that sample from the input distribution? Virtually all other work on neural sampling that I am aware of (e.g., from Maass Lab, Lengyel Lab, etc.) start from a single goal function for all connections that somehow quantifies the difference of network dynamics from the target distribution. In the presented work the authors specify different goal functions for the different weights, which does not make clear how the desired network dynamics are ultimately achieved.

This becomes especially evident looking at the two different recurrent connections (M and G). M minimizes the difference between network activity f and recurrent prediction DKL[f|phi(My)], but why is this alone not enough to ensure a good sampling? G minimizes the squared error [f-phi(Gy)]^2, but what does that mean? The problem is that the goal functions are self-consistent in the sense that both f and phi(Gy) depend on G, which makes an interpretation very difficult. Ultimately it's easier to interpret this by looking at the plasticity rule and see that it leads to a balance. For G the authors furthermore actually ignore the derived plasticity rule and switch to a rule similar to the one for M, meaning that the actual goal function for G is also something like DKL[f|phi(Gy)]. Overall, an overarching optimization goal for the entire network is missing, which makes the interpretation very difficult. I understand that this might be very difficult to provide at this stage, but the authors should at least point out this shortcoming as an open question for the proposed framework.

Regarding the relation to previous work the authors have provided a lot more detailed discussion, which very much clears up the contributions and novel ideas in their work. Still, there are some claims that are not consistent with the literature. Especially, in lines 767 ff. the authors state that Kappel et al "assumed plasticity only at recurrent synapses projecting onto the excitatory neurons. In addition, unlike our model, the cell assembly memberships need to be preconfigured in the [...] model." This is not correct, as Kappel et al learn both the feed-forward and recurrent connections, hence the main difference is that in Kappel et al sampling is sequential and not random. This is why I mentioned this work in the first review, as it speaks against the authors claims of novelty (719 ff.), which should be adjusted accordingly.

---

## [Referee Report · Reviewer #2 (Public review)]

Summary:

The paper reconsiders the formation of Hebbian-type assemblies, with their spontaneous reactivation representing the statistics of the sensory inputs, in the light of predictive synaptic plasticity. It convincingly shows that not all plasticity rules can be predictive in the narrow sense. While plasticity for the excitatory synapses (the forward projecting and recurrent ones) are predictive, two types of plasticity in the recurrent inhibition is required: a homeostatic and competitive one.

Details:

Besides the excitatory forward and recurrent connections that are learned based on predictive synaptic plasticity, two types of inhibitory plasticity are considered. A first type of inhibition is homeostatic and roughly balances excitation within the cell assemblies. Plasticity in this type 1 inhibition is also predictive, analogous to the plasticity of the excitatory synapses. However, plasticity in type 2 inhibition is competitive and has a switched sign. Both types of inhibitory plasticity, the predictive (homeostatic) and the anti-predictive (competitive) one, work together with the predictive excitatory plasticity to form cell assemblies representing sensory stimuli. Only if the two types of homeostatic and competitive inhibitory plasticity are present, will the spontaneous replay of the assemblies reflect the statistics of the stimulus presentation.

Critical review:

The simulations include Dale's law, making them more biologically realistic. The paper emphasizes predictive plasticity and introduces type 1 inhibitory plasticity that, by construction, tries to fully explain away the excitatory input. In the absence of external inputs, however, due to the symmetry between the excitatory and inhibitory-type-1 plasticity rules, excitation and inhibition tend to fully cancel each other. Multiple options may solve the dilemma:

(1) As other predictive dendritic plasticity models assume, the presynaptic source for recurrent inhibition is typically less informative than the presynaptic source of excitation, so that inhibition is not able to fully explain away excitation.

(2) Beside the inhibitory predictive plasticity that mirrors the analogous excitatory predictive plasticity, and additional competitive plasticity can be introduced.

The paper chooses solution (2) and suggests and additional inhibitory recurrent pathway that is not predictive, but instead anti-predictive with a reversed sign. The combination of the two types of inhibitory plasticities lead to a stable formation of cell assemblies. The stable target activity of the plasticity rules in a memory recall is not anymore 0, as it would be with only type-1-inhibitory plasticity.

Instead, the target activity of plasticity is now enhanced within a winning assembly, and also positive but reduced in the loosing assemblies.

---

## [Referee Report · Reviewer #3 (Public review)]

Summary:

The work shows how learned assembly structure and its influence on replay during spontaneous activity can reflect the statistics of stimulus input. In particular, stimuli that are more frequent during training elicit stronger wiring and more frequent activation during replay. Past works (Litwin-Kumar and Doiron, 2014; Zenke et al., 2015) have not addressed this specific question, as classic homeostatic mechanisms forced activity to be similar across all assemblies. Here, the authors use a dynamic gain and threshold mechanism to circumnavigate this issue and link this mechanism to a cellular monitoring of membrane potential history.

Strengths:

(1) This is an interesting advance, and the authors link this to experimental work in sensory learning in environments with non-uniform stimulus probabilities.

(2) The authors consider their mechanism in a variety of models of increasing complexity (simple stimuli, complex stimuli; ignoring Dale's law, incorporating Dale's law).

(3) Links a cellular mechanism of internal gain control (their variable h) to assembly formation and the non-uniformity of spontaneous replay activity. Offers a promise of relating cellular and synaptic plasticity mechanisms under a common goal of assembly formation.

Weaknesses:

(1) However, while the manuscript does show that assembly wiring does follow stimulus likelihood, it is not clear how the assembly specific statistics of h reflect these likelihoods. I find this to be a key issue.

(2) The authors model does take advantage of the sigmoidal transfer function, and after learning an assembly is either fully active or near fully silent (Fig. 2a). This somewhat artificial saturation may be the reason that classic homeostasis is not required, since runaway activity is not as damaging to network activity.

(3) Classic mechanisms of homeostatic regulation (synaptic scaling, inhibitory plasticity) try to ensure that firing rates match a target rate (on average). If the target rate is the same for all neurons then having elevated firing rates for one assembly compared to others during spontaneous activity would be difficult. If these homeostatic mechanisms were incorporated, how would they permit the elevated firing rates for assemblies that represent more likely stimuli?

---

## [Author Response]

The following is the authors’ response to the previous reviews.

**Public Reviews:**

**Reviewer #1 (Public Review):**
In their manuscript, the authors propose a learning scheme to enable spiking neurons to learn the appearance probability of inputs to the network. To this end, the neurons rely on error-based plasticity rules for feedforward and recurrent connections. The authors show that this enables the networks to spontaneously sample assembly activations according to the occurrence probability of the input patterns they respond to. They also show that the learning scheme could explain biases in decision-making, as observed in monkey experiments. While the task of neural sampling has been solved before in other models, the novelty here is the proposal that the main drivers of sampling are within-assembly connections, and not between-assembly (Markov chains) connections as in previous models. This could provide a new understanding of how spontaneous activity in the cortex is shaped by synaptic plasticity.The manuscript is well written and the results are presented in a clear and understandable way. The main results are convincing, concerning the spontaneous firing rate dependence of assemblies on input probability, as well as the replication of biases in the decision-making experiment. Nevertheless, the manuscript and model leave open several important questions. The main problem is the unclarity, both in theory and intuitively, of how the sampling exactly works. This also makes it difficult to assess the claims of novelty the authors make, as it is not clear how their work relates to previous models of neural sampling.

We agree with the reviewer that our previous manuscript was not clear regarding the mechanism of the model. We have performed additional simulations and included a derivation of the learning rule to address this, which we explain below.

Regarding the unclarity of the sampling mechanism, the authors state that withinassembly excitatory connections are responsible for activating the neurons according to stimulus probability. However, the intuition for this process is not made clear anywhere in the manuscript. How do the recurrent connections lead to the observed effect of sampling? How exactly do assemblies form from feedforward plasticity? This intuitive unclarity is accompanied by a lack of formal justification for the plasticity rules. The authors refer to a previous publication from the same lab, but it is difficult to connect these previous results and derivations to the current manuscript. The manuscript should include a clear derivation of the learning rules, as well as an (ideally formal) intuition of how this leads to the sampling dynamics in the simulation.

We have included a derivation of our plasticity rules in lines 871-919 in the revised manuscript. Consistent with our claim that predictive plasticity updates the feedforward and the recurrent synapses to predict output firing rates, we have shown that the corresponding cost function measures the discrepancy among the recurrent prediction, feedforward prediction, and the output firing rate. The resultant feedforward plasticity is the same with our previous rule (Asabuki and Fukai, 2020), which segments the salient patterns embedded in the input sequence. The recurrent plasticity rule suggests that the recurrent prediction learns the statistical model of the evoked activity, enabling the network to replay the learned internal model.

Similarly, for the inhibitory plasticity, we defined a cost function that evaluates the difference between the firing rate and inhibitory potential within each neuron. This rule is crucial for maintaining balanced network dynamics. See our response below for more details on the role of inhibitory plasticity.

Some of the model details should furthermore be cleared up. First, recurrent connections transmit signals instantaneously, which is implausible. Is this required, would the network dynamics change significantly if, e.g., excitation arrives slightly delayed? Second, why is the homeostasis on h required for replay? The authors show that without it the probabilities of sampling are not matched, but it is not clear why, nor how homeostasis prevents this. Third, G and M have the same plasticity rule except for G being confined to positive values, but there is no formal justification given for this quite unusual rule. The authors should clearly justify (ideally formally) the introduction of these inhibitory weights G, which is also where the manuscript deviates from their previous 2020 work. My feeling is that inhibitory weights have to be constrained in the current model because they have a different goal (decorrelation, not prediction) and thus should operate with a completely different plasticity mechanism. The current manuscript doesn't address this, as there is no overall formal justification for the learning algorithm.

First, while the reviewer's suggestion to test with delayed excitation is intriguing and crucial for a more biologically detailed spiking neuron model, we have chosen to maintain the current model configuration. Our use of Poisson spiking neurons, which generate spikes based on instantaneous firing rates, does not heavily depend on precise spike timing information. Therefore, to preserve the simplicity of our results, we kept the model unchanged.

Second, we agree that our previous claim regarding the importance of the memory trace h for sampling may have been confusing. As shown in Supplementary Figure 7b in the revised manuscript, when we eliminated the dynamics of the memory trace, sampling performance did indeed decrease. However, we also observed that the assembly activity ratio continued to show a linear relationship with stimulus probabilities. Based on these findings, we have revised our claim in the manuscript to clarify that the memory trace is primarily critical for firing rate homeostasis, rather than directly influencing sampling within the learned network. We have explained this in ll. 446-448 in the revised manuscript.

Third, we explored a new architecture where all recurrent connections are either exclusively excitatory or inhibitory, keeping their sign throughout the learning process. This change addresses the reviewer's concern about our initial assumption that only the inhibitory connection G was constrained to non-negative values. We found that inhibition plays a crucial role in decorrelation and prediction, helping activate specific assemblies through competition while preventing runaway excitation within active assemblies. We have explained this in ll.560-593 in the revised manuscript.

Finally, the authors should make the relation to previous models of sampling and error-based plasticity more clear. Since there is no formal derivation of the sampling dynamics, it is difficult to assess how they differ exactly from previous (Markov-based) approaches, which should be made more precise. Especially, it would be important to have concrete (ideally experimentally testable) predictions on how these two ideas differ. As a side note, especially in the introduction (line 90), this unclarity about the sampling made it difficult to understand the contrast to Markovian transition models.

As the reviewer pointed out, previous computational models have demonstrated that recurrent networks with Hebbian-like plasticity can learn appropriate Markovian statistics (Kappel et al., 2014; Asabuki and Clopath, 2024). However, our model differs conceptually from these previous models. While Kappel et al. showed that STDP in winner-take-all circuits can approximate online learning of hidden Markov models (HMMs), a key difference with our model is that their neural representations acquire sequences using Markovian sampling dynamics, whereas our model does not depend on such ordered sampling. Specifically, in their model, sequential sampling arises from learned structures in the off-diagonal elements of the recurrent connections (i.e., between-assembly connections). In contrast, our network learns to stochastically generate recurrent cell assemblies by relying solely on within-assembly connections. A similar argument can be made for Asabuki and Clopath paper as well. Further, while our model introduced plasticity rule for all types of connections, Asabuki and Clopath paper introduced plasticity for recurrent synapses projecting on the excitatory neurons only and the cell assembly memberships were preconfigured unlike our model. We have added additional clarifying sentences in ll. 757-772 of the revised manuscript to elaborate on this point.

There are also several related models that have not been mentioned and should be discussed. In 663 ff. the authors discuss the contributions of their model which they claim are novel, but in Kappel et al (STDP Installs in Winner-Take-All Circuits an Online Approximation to Hidden Markov Model Learning) similar elements seem to exist as well, and the difference should be clarified. There is also a range of other models with lateral inhibition that make use of error-based plasticity (most recently reviewed in Mikulasch et al, Where is the error? Hierarchical predictive coding through dendritic error computation), and it should be discussed how the proposed model differs from these.

We have clarified the difference from previously proposed recurrent network model to perform Markovian sampling. Please see our reply above.

We have also included additional sentence in ll. 704-709 in the revised manuscript to discuss how our model differs from similar predictive learning models: “It should be noted that while several network models that perform errorbased computations like ours exploit only inhibitory recurrent plasticity (Mikulasch et al., 2021; Mackwood et al., 2021; Hertäg and Clopath., 2022; Mikulasch et al., 2023), our model learns the structured spontaneous activity to reproduce the evoked statistics by modifying both excitatory and inhibitory recurrent connections.”

**Reviewer #2 (Public Review):**
Summary:The paper considers a recurrent network with neurons driven by external input. During the external stimulation predictive synaptic plasticity adapts the forward and recurrent weights. It is shown that after the presentation of constant stimuli, the network spontaneously samples the states imposed by these stimuli. The probability of sampling stimulus x^(i) is proportional to the relative frequency of presenting stimulus x^(i) among all stimuli i=1,..., 5.Methods:Neuronal dynamics:For the main simulation (Figure 3), the network had 500 neurons, and 5 nonoverlapping stimuli with each activating 100 different neurons where presented. The voltage u of the neurons is driven by the forward weights W via input rates x, the inhibitory recurrent weights G, are restricted to have non-negative weights (Dale's law), and the other recurrent weights M had no sign-restrictions. Neurons were spiking with an instantaneous Poisson firing rate, and each spike-triggered an exponentially decaying postsynaptic voltage deflection. Neglecting time constants of the postsynaptic responses, the expected postsynaptic voltage reads (in vectorial form) asu = W x + (M - G) f (Eq. 5)where f = ; phi(u) represents the instantaneous Poisson rate, and phi a sigmoidal nonlinearity. The rate f is only an approximation (symbolized by = ;) of phi(u) since an additional regularization variable h enters (taken up in Point 4 below). The initialisation of W and M is Gaussian with mean 0 and variance 1/sqrt(N), N the number of neurons in the network. The initial entries of G are all set to 1/sqrt(N).Predictive synaptic plasticity:The 3 types of synapses were each adapted so that they individually predict the postsynaptic firing rate f, in matrix formΔW ≈ (f - phi(W x)) x^TΔM ≈ (f - phi(M f)) f^TΔG ≈ (f - phi(M f)) f^T but confined to non-negative values of G (Dale's law).The ^T tells us to take the transpose, and the ≈ again refers to the fact that the ϕ entering in the learning rule is not exactly the ϕ determining the rate, only up to the regularization (see Point 4).Main formal result:As the authors explain, the forward weight W and the unconstrained weight M develop such that, in expectations,f = ; phi(W x) = ; phi(M f) = ; phi(G f) ,consistent with the above plasticity rules. Some elements of M remain negative. In this final state, the network displays the behaviour as explained in the summary.Major issues:Point 1: Conceptual inconsistencyThe main results seem to arise from unilaterally applying Dale's law only to the inhibitory recurrent synapses G, but not to the excitatory recurrent synapses M.In fact, if the same non-negativity restriction were also imposed on M (as it is on G), then their learning rules would become identical, likely leading to M=G. But in this case, the network becomes purely forward, u = W x, and no spontaneous recall would arise. Of course, this should be checked in simulations.Because Dale's law was only applied to G, however, M and G cannot become equal, and the remaining differences seem to cause the effect.Predictive learning rules are certainly powerful, and it is reasonable to consider the same type of error-correcting predictive learning rule, for instance for different dendritic branches that both should predict the somatic activity. Or one may postulate the same type of error-correcting predictive plasticity for inhibitory and excitatory synapses, but then the presynaptic neurons should not be identical, as it is assumed here. Both these types of error-correcting and error-forming learning rules for same-branches and inhibitory/excitatory inputs have been considered already (but with inhibitory input being itself restricted to local input, for instance).

The model presented above lacked biological plausibility in several key aspects. Specifically, we assumed that the recurrent connection M could change sign through plasticity and be either excitatory or inhibitory, while the inhibitory connection G was restricted to being inhibitory only. This initial setting does not reflect the biological constraint that synapses typically maintain a consistent excitatory or inhibitory type. Furthermore, due to this unconstrained recurrent connectivity M, the original model had two types of inhibitory connections (i.e., the negative part of M and the inhibitory connection G) without providing a clear computational role for each type of inhibition.

To address these limitations and to understand the role of the two types of inhibition, we explored a new architecture where all recurrent connections are either exclusively excitatory or inhibitory, keeping their sign throughout the learning process. This change addresses the reviewer's concern about our initial assumption that only the inhibitory connection G was constrained to non-negative values. We found that inhibition plays a crucial role in prediction and decorrelation, helping activate specific assemblies through competition while preventing runaway excitation within active assemblies. We have explained this in ll. 561593 in the revised manuscript.

Point 2: Main result as an artefact of an inconsistently applied Dale's law?The main result shows that the probability of a spontaneous recall for the 5 nonoverlapping stimuli is proportional to the relative time the stimulus was presented. This is roughly explained as follows: each stimulus pushes the activity from 0 up towards f = ; phi(W x) by the learning rule (roughly). Because the mean weights W are initialized to 0, a stimulus that is presented longer will have more time to push W up so that positive firing rates are reached (assuming x is non-negative). The recurrent weights M learn to reproduce these firing rates too, while the plasticity in G tries to prevent that (by its negative sign, but with the restriction to non-negative values). Stimuli that are presented more often, on average, will have more time to reach the positive target and hence will form a stronger and wider attractor. In spontaneous recall, the size of the attractor reflects the time of the stimulus presentation. This mechanism so far is fine, but the only problem is that it is based on restricting G, but not M, to non-negative values.

As mentioned above, we have included an additional simulation where all weights are non-negative. We have demonstrated the new results in Figure 6 before presenting the two-population model in the revised manuscript (Figure 7), so that readers can follow the importance of two pathways of inhibitory connections.

Point 3: Comparison of rates between stimulation and recall.The firing rates with external stimulations will be considerably larger than during replay (unless the rates are saturated).This is a prediction that should be tested in simulations. In fact, since the voltage roughly reads as u = W x + (M - G) f, and the learning rules are such that eventually M = ; G, the recurrences roughly cancel and the voltage is mainly driven by the external input x. In the state of spontaneous activity without external drive, one has u = (M - G) f , and this should generate considerably smaller instantaneous rates f = ; phi(u) than in the case of the feedforward drive (unless f is in both cases at the upper or lower ceiling of phi). This is a prediction that can also be tested.Because the figures mostly show activity ratios or normalized activities, it was not possible for me to check this hypothesis with the current figures. So please show non-normalized activities for comparing stimulation and recall for the same patterns.

We agree with the reviewer that the activity levels of spontaneous and induced activity should be compared. We have shown the distributions of activity level of these activities in our new Figure 2d. As expected, we found that the evoked activity showed stronger activity compared to the spontaneous activity.

Point 4: Unclear definition of the variable h.The formal definition of h = hi is given by (suppressing here the neuron index i and the h-index of tau)tau dh/dt = -h if h>u, (Eq. 10) h = u otherwise.But if it is only Equation 10 (nothing else is said), h will always become equal to u, or will vanish, i.e. either h=u or h=0 after some initial transient. In fact, as soon as h>u, h is decaying to 0 according to the first line. If u is >0, then it stops at u=h according to the second line. No reason to change h=u further. If u<=0 while h>u, then h is converging to 0 according to the first line and will stay there. I guess the authors had issues with the recurrent spiking simulations and tried to fix this with some regularization. However as presented, it does not become clear how their regulation works.

We apologize for the reviewer that our definition of h was unclear. As the reviewer pointed out, since the memory trace is always positive and larger than (or equal to) the membrane potential, it is possible that the membrane potential becomes always negative and the memory trace reach to 0 constantly. However, since the network is always balanced between excitatory and inhibitory inputs, and it does not happen that the membrane potential always diverges negatively. In fact, we trained without any manipulations other than the memory trace described in the manuscript, and the network was able to learn the assembly structure stably.

BTW: In Eq. 11 the authors set the gain beta to beta = beta0/h which could become infinite and, putatively more problematic, negative, depending on the value of h. Maybe some remark would convince a reader that no issues emerge from this.

We have mentioned in ll. 864-866 in the revised manuscript that no issues emerge from the slope parameter.

Added from discussions with the editor and the other reviewers:Thanks for alerting me to this Supplementary Figure 8. Yes, it looks like the authors did apply there Dale's law for both the excitatory and inhibitory synapses. Yet, they also introduced two types of inhibitory pathways converging both to the excitatory and inhibitory neurons. For me, this is a confirmation that applying Dale's law to both excitatory and inhibitory synapses, with identical learning rules as explained in the main part of the paper, does not work.Adding such two pathways is a strong change from the original model as introduced before, and based on which all the Figures in the main text are based. Supplementary Figure 8 should come with an analysis of why a single inhibitory pathway does not work. I guess I gave the reason in my Points 1-3. Some form of symmetry breaking between the recurrent excitation and recurrent inhibition is required so that, eventually, the recurrent excitatory connection will dominate.Making the inhibitory plasticity less expressive by applying Dale's law to only those inhibitory synapses seems to be the answer chosen in the Figures of the main text (but then the criticism of unilaterally applying Dale's law).Applying Dale's law to both types of synapses, but dividing the labor of inhibition into two strictly separate and asymmetric pathways, and hence asymmetric development of excitatory and inhibitory weights, seems to be another option. However, introducing such two separate inhibitory pathways, just to rescue the fact that Dale's law is applied to both types of synapses, is a bold assumption. Is there some biological evidence of such two pathways in the inhibitory, but not the excitatory connections? And what is the computational reasoning to have such a separation, apart from some form of symmetry breaking between excitation and inhibition? I guess, simpler solutions could be found, for instance by breaking the symmetry between the plasticity rules for the excitatory and inhibitory neurons. All these questions, in my view, need to be addressed to give some insights into why the simulations do work.

The reviewer’s intuition is correct. To effectively learn cell assembly structures and replay their activities, our model indeed requires two types of inhibitory connections. Please refer to our response above for further details.

Overall, Supplementary Figure 8 seems to me too important to be deferred to the Supplement. The reasoning behind the two inhibitory pathways should appear more prominently in the main text. Without this, important questions remain. For instance, when thinking in a rate-based framework, the two inhibitory pathways twice try to explain the somatic firing rate away. Doesn't this lead to a too strong inhibition? Can some steady state with a positive firing rate caused by the recurrence, in the absence of an external drive, be proven? The argument must include the separation into Path 1 and Path 2. So far, this reasoning has not been entered.In fact, it might be that, in a spiking implementation, some sparse spikes will survive. I wonder whether at least some of these spikes survive because of the other rescuing construction with the dynamic variable h (Equation 10, which is not transparent, and that is not taken up in the reasoning either, see my Point 4)Perhaps it is helpful for the authors to add this text in the reply to them.

We have moved the former Supplemental Figure 8 to the main Figure 7. Please see our response above about the role of dual inhibitory connection types.

**Reviewer #3 (Public Review):**
Summary:The work shows how learned assembly structure and its influence on replay during spontaneous activity can reflect the statistics of stimulus input. In particular, stimuli that are more frequent during training elicit stronger wiring and more frequent activation during replay. Past works (Litwin-Kumar and Doiron, 2014; Zenke et al., 2015) have not addressed this specific question, as classic homeostatic mechanisms forced activity to be similar across all assemblies. Here, the authors use a dynamic gain and threshold mechanism to circumnavigate this issue and link this mechanism to cellular monitoring of membrane potential history.Strengths:(1) This is an interesting advance, and the authors link this to experimental work in sensory learning in environments with non-uniform stimulus probabilities.(2) The authors consider their mechanism in a variety of models of increasing complexity (simple stimuli, complex stimuli; ignoring Dale's law, incorporating Dale's law).(3) Links a cellular mechanism of internal gain control (their variable h) to assembly formation and the non-uniformity of spontaneous replay activity. Offers a promise of relating cellular and synaptic plasticity mechanisms under a common goal of assembly formation.Weaknesses:(1) However, while the manuscript does show that assembly wiring does follow stimulus likelihood, it is not clear how the assembly-specific statistics of h reflect these likelihoods. I find this to be a key issue.

We agree that our previous claim regarding the importance of the memory trace h for sampling may have been confusing. As shown in Supplementary Figure 7b, when we eliminated the dynamics of the memory trace, sampling performance did indeed decrease. However, we also observed that the assembly activity ratio continued to show a linear relationship with stimulus probabilities. Based on these findings, we revised our claim in the manuscript to clarify that the memory trace is primarily critical for learning to avoid trivial solutions, rather than directly influencing sampling within the learned network. We have explained this in ll. 446-448 in the revised manuscript.

(2) The authors' model does take advantage of the sigmoidal transfer function, and after learning an assembly is either fully active or nearly fully silent (Figure 2a). This somewhat artificial saturation may be the reason that classic homeostasis is not required since runaway activity is not as damaging to network activity.

The reviewer's intuition is correct. The saturating nonlinearity is important for the network to form stable assembly structures. We have added an additional sentence in ll. 866-868 to mention this.

(3) Classic mechanisms of homeostatic regulation (synaptic scaling, inhibitory plasticity) try to ensure that firing rates match a target rate (on average). If the target rate is the same for all neurons then having elevated firing rates for one assembly compared to others during spontaneous activity would be difficult. If these homeostatic mechanisms were incorporated, how would they permit the elevated firing rates for assemblies that represent more likely stimuli?

LIF neurons may solve this problem by utilizing spike-timing statistics.

**Recommendations for the authors:**

**Reviewer #1 (Recommendations For The Authors):**
Minor issues:Figure 1: It would be helpful to display the equation for output rate here as well.

We have included the equation in the revised Figure 1a.

Figure 3c: Typo "indivisual neurons".

We have modified the typo. We thank the reviewer for their careful review.

Line 325: Do you mean Figure 3f,g?

We repeated the task with different numbers of stimuli in Supplementary Figure 1c,d.

Line 398: Winner-take-all can be misunderstood, as it typically stands for competition in inference, not in learning.

We have rephrased it as “unstable dynamics” in l. 400

Line 429: Are intra-assembly and within-assembly the same? If so please use consistent terminology.

We have made the terminology consistent.

Line 792 ff.: Please mention that (t) was left away.

We have included a sentence to mention it in ll. 847-848 in the revised manuscript.

Line 817: Should u_i be v_i?

We have modified the term.

Methods: What is the value of tau_h?

We have used 𝜏! = 10 s, which is mentioned in l. 853